# PARP7 as a new target for activating anti-tumor immunity in cancer

Katerina Popova [ID] [1,2,3], Johannes Benedum [ID] [1,2,4,5,6], Magdalena Engl [ID] [1,2,4,5,6], Carola Lütgendorf-Caucig [ID] [3,7], Piero Fossati [3,8], Joachim Widder [ID] [1,2], Klaus Podar [8,9] & Dea Slade [ID] [1,2,3,4,5] [✉]

## Abstract

**ADP-ribosyl transferases (ARTs) are a family of enzymes which catalyze the addition of a chain (PARylation) or a single moiety (MARylation) of ADP-ribose to their substrates. PARP7 is a mono-ADP-ribosyl transferase (mono-ART) which has recently gained attention due to its emerging role as a negative regulator of the type I interferon (IFN-I) and nuclear receptor signaling, and due to its aberrant expression in cancer, contributing to disease progression and immune evasion. PARP7-mediated ADP-ribosylation can differentially affect protein stability. On the one hand, PARP7-mediated ADP-ribosylation of the transcription factor FRA1 protects it from proteosomal degradation and thereby supports its function in negatively regulating IRF1 and the expression of apoptosis and immune signaling genes. On the other hand, PARP7-mediated ADP-ribosylation of aryl hydrocarbon receptor (AHR) and estrogen receptor (ER) marks them for proteosomal degradation. PARP7 also ADP-ribosylates the ligand-bound androgen receptor (AR), which is recognized by DTX3L-PARP9 that modulate the AR transcriptional activity. In this review, we discuss PARP7 enzymatic properties, biological functions and known substrates, its role in various cancers, and its targeting by specific inhibitors.**

**Keywords** PARP7; ADP-ribosylation; Type I Interferon Response; Anti-tumor Immunity; Nuclear Receptors
**Subject Categories** Cancer; Immunology

## Introduction

Post-translational protein modifications (PTMs) enable a fast response to internal and external stimuli. PTMs can be reversible, such as the addition of small chemical groups (e.g., phosphorylation), or irreversible, such as proteolytic cleavage. ADP-ribosylation is a reversible PTM, associated with a multitude of cellular processes such as metabolic regulation, DNA damage repair, transcriptional regulation, and signal transduction. It is deposited by the enzymes of the ADP-ribosyl transferase (ARTs) family (Lüscher et al, 2022). A few members of this class of proteins (PARP1, PARP2, TNKS1, TNKS2) have the ability to attach a chain of ADP-ribose (ADPr) to their substrates – a process referred to as poly-ADP-ribosylation (PARylation). The majority of ARTs are only able to mono-ADP-ribosylate (MARylate) their targets. One such mono-ART that has garnered a lot of attention in recent years is PARP7, also known as TCDD-inducible poly-ADP-ribose polymerase (TiPARP). PARP7 has emerged as a key regulator in innate immune signaling and plays a crucial role in cancer immunity. More specifically, PARP7 acts as a negative regulator of the type I interferon (IFN-I) signaling on multiple levels, prompting efforts to develop highly selective PARP7 inhibitors as a new therapeutic strategy. This review aims to collect and discuss the currently available literature on PARP7, with the focus on its enzymatic properties, substrates, functions, and clinical relevance.

## Enzymatic properties of PARP7

ARTs are a family of 17 multi-domain enzymes, which modify target proteins by catalyzing the transfer of a single (MARylation) or multiple (PARylation) moieties of ADP-ribose (ADPr) from nicotinamide adenine dinucleotide (NAD$^+$) to their target proteins. Some ARTs can also modify terminal phosphates and bases of nucleic acids (Munnur et al, 2019; Musheev et al, 2022; Talhaoui et al, 2016). ADP-ribosylation can be reversed by ADP-ribosyl hydrolases, which can specifically degrade MAR (MACROD1, MACROD2, ARH1), PAR (PARG), or both (ARH3, TARG) (Feijs and Žaja, 2022). The catalytic domain of PARylating ARTs, such as PARP1, PARP2, TNKS1 (PARP5a), and TNKS2 (PARP5b), contains the H-Y-E motif, where histidine and tyrosine are needed for the binding of the NAD$^+$ molecule, whereas glutamate is important for the generation of PAR chains (Marsischky et al, 1995; Papini et al, 1989). However, despite

[1]Department of Radiation Oncology, Medical University of Vienna, Währinger Gürtel 18-20, 1090 Vienna, Austria. [2]Comprehensive Cancer Center, Medical University of Vienna, Spitalgasse 23, 1090 Vienna, Austria. [3]MedAustron Ion Therapy Center, Wiener Neustadt, Austria. [4]Max Perutz Labs, Vienna Biocenter Campus (VBC), Dr.-Bohr-Gasse 9, 1030 Vienna, Austria. [5]Medical University of Vienna, Center for Medical Biochemistry, Dr.-Bohr-Gasse 9, 1030 Vienna, Austria. [6]Vienna Biocenter PhD Program, a Doctoral School of the University of Vienna and the Medical University of Vienna, 1030 Vienna, Austria. [7]Heidelberg Ion-Beam Therapy Center (HIT), Department of Radiation Oncology, Heidelberg University Hospital, Heidelberg, Germany. [8]Division of Molecular Oncology and Hematology, Department of Basic and Translational Oncology, Karl Landsteiner University of Health Sciences, Krems an der Donau, Austria. [9]Division of Internal Medicine 2, University Hospital Krems, Krems and der Donau, Austria. [✉]E-mail: dea.slade@maxperutzlabs.ac.at

**Glossary**

**AHR**
aryl hydrocarbon receptor, a ligand-dependent transcription factor, which functions as an environmental sensor and induces the expression of drug-metabolizing enzymes

**ART**
ADP-ribosyl transferase, transfers ADP-ribose from NAD$^+$ to substrates

**cGAS-STING pathway**
a cytosolic DNA sensing pathway, which activates the innate immune response in the case of infection or if DNA is present in the cytosol due to DNA damage (e.g., mitochondrial damage, micronuclei, anaphase bridges). Upon sensing cytosolic dsDNA, cGAS produces the second messenger cGAMP, which activates STING. STING serves as a signaling platform for the recruitment and activation of TBK1, IRF3 and NF-κB for the transcriptional activation of type I interferons.

**IFN-I**
type I interferon, produced upon activation of cGAS-STING or RIG-I-MAVS pathways. Upon binding to interferon-α/β receptor (IFNAR), it induces JAK/STAT signaling, resulting in activation of STAT transcription factors and expression of interferon-stimulated genes (ISGs). ISGs are chemokines and cytokines that attract and activate immune cells.

**Innate immunity**
The first line of defense against infection, which relies on physical barriers and recruits immune cells to infection sites by producing cytokines and chemokines. It is a rapid and non-specific response, which can activate the adaptive immune response by antigen presentation.

**MARylation**
mono-ADP-ribosylation, a single unit of ADP-ribose attached to an amino acid or a nucleic acid

**NF-κB pathway**
a signaling pathway that controls cell proliferation and immune response by inducing gene expression. Under normal conditions, NF-κB is kept inactive in the cytoplasm by an inhibitor (IκB). Under stress conditions, the inhibitor is degraded and NF-κB translocates into the nucleus to activate the expression of inflammatory cytokines and proliferation genes.

**PARylation**
poly-ADP-ribosylation, multiple units of ADP-ribose attached to an amino acid or a nucleic acid

PARP3 and PARP4 also having this signature motif, they are unable to PARylate their substrates, implying that other residues also play a role in determining the catalytic activity (Vyas et al, 2014).

The catalytic triad of MARylating ARTs contains tyrosine, leucine, or isoleucine instead of the glutamate residue (Vyas et al, 2014). PARP7 belongs to the latter category of ARTs and alongside its catalytic domain, which contains an H-Y-I motif, it possesses a CCCH-type zinc-finger domain and two WWE domains (tryptophan-tryptophan-glutamate) (Fig. 1A), only one of which was shown to be functional as an ADP-ribose binding domain (WWE2). Human PARP7 is located on chromosome 3 and its full-length protein has a molecular weight of 76 kDa. Although the structure of PARP7 is not available, AlphaFold2 predicted interactions between the C-terminal region of the catalytic domain of PARP7 and the WWE1 domain (Suskiewicz et al, 2023) (Fig. 1B). With regard to MARylation, varying amino acids can act as ADPr acceptors. Some of the identified ones thus far are the basic amino acid arginine, the acidic amino acids glutamate and aspartate, as well as tyrosine and cysteine (Buch-Larsen et al, 2020; Larsen et al, 2018; Leslie Pedrioli et al, 2018; Rodriguez et al, 2021). Using a chemical genetics approach, Rodriguez et al demonstrated that PARP7 mainly MARylates cysteines (Rodriguez et al, 2021). PARP7 is auto-MARylated on C39 (Gomez et al, 2018), C543 and C552, which are located in its catalytic domain (Rodriguez et al, 2021), and C100 and C439 (Yang et al, 2021) (Table 1; Fig. 1B). AlphaFold3 (Abramson et al, 2024) predicts the structures of PARP7 bound to its known substrates tubulin (TBA4A), hypoxia-inducible factor 1-α (HIF-1α) and nuclear factor (NF)-κB RelA (p65) subunit, whereby WWE2 domain positions the substrate for MARylation by the catalytic domain (Fig. 1C–E). Flexibility between the WWE1 domain and the catalytic domain may explain the presence of additional ADP-ribosylation sites away from the catalytic domain in the AlphaFold3 model of PARP7-TBA4A (Fig. 1C). While MARylation sites for HIF-1α and RelA have not been reported yet, AlphaFold3 models indicate putative MARylation sites, which are exposed towards the catalytic region of PARP7 (Fig. 1D,E).

# PARP7 biological functions and substrates

PARP7 localizes mainly in the nucleus, which requires the N-terminal nuclear localization sequence (NLS) region and an intact zinc-finger domain CCCH (Gomez et al, 2018) (Fig. 1A,B). PARP7 localizes also in the cytosol, as observed in HeLa and MCF7 cells (MacPherson et al, 2014; Vyas et al, 2013). Furthermore, following infection with Sindbis virus (SINV), PARP7 accumulates in the cytosol of mouse embryonic fibroblasts (MEFs) (Kozaki et al, 2017). This translocation is facilitated by oxidation of the nucleoporin complex via reactive oxygen species (ROS) from damaged mitochondria, which is mediated by the BCL2 family members BAX and BAK1. PARP7 nuclear localization is crucial for its function as a modulator of nuclear receptor-mediated transcription and IFN-I signaling, while its cytosolic localization may contribute to the regulation of IFN-I signaling and microtubules, which will be described in detail below.

## PARP7 in nuclear receptor signaling

PARP7 expression is regulated by different transcription factors, and PARP7 in turn regulates the transcriptional activity of different nuclear receptors. PARP7 was first identified as a target gene of the aryl hydrocarbon receptor (AHR), which acts as a ligand-activated transcription factor (Diani-Moore et al, 2010) (Fig. 2). Upon the binding of ligands such as aromatic hydrocarbons or tryptophan metabolites, AHR is activated and translocates into the nucleus where it binds together with aryl hydrocarbon receptor nuclear translocator (ARNT) to the AHR response elements (AHREs) in the regulatory regions of genes such as cytochrome P450 1A1 (*CYP1A1*), *CYP1B1*, AHR repressor (*AHRR*) and PARP7 (*TIPARP*), leading to

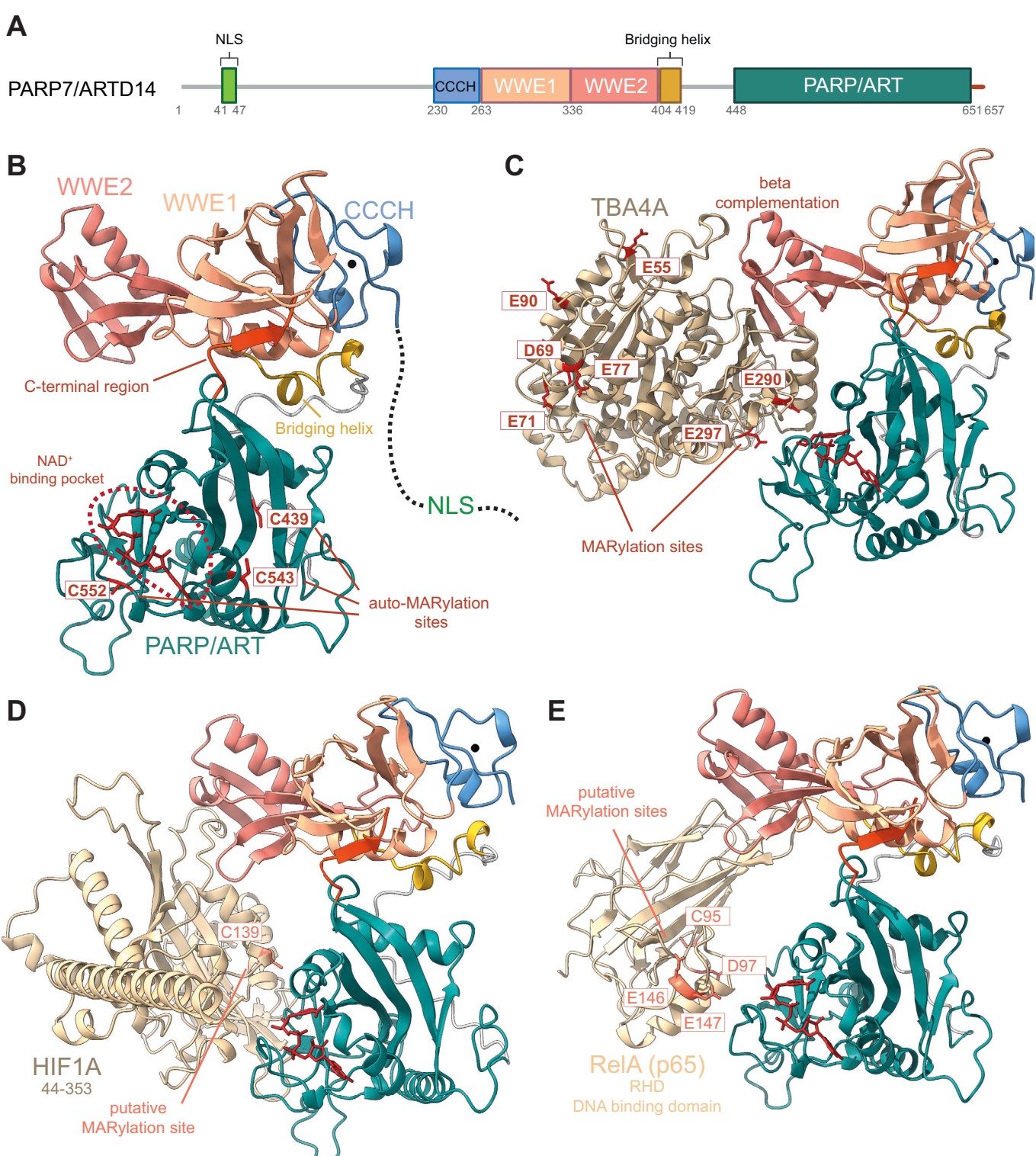

**Figure 1. PARP7 domain architecture, structural model, and substrate recognition.**

(A) Schematic representation of PARP7 domains. PARP7 is a mono-ADP-ribosyl transferase, which is comprised of the catalytic (PARP/ART) domain, containing an H-Y-I motif, two tryptophan-tryptophan-glutamate (WWE) domains, one CCCH-type zinc-finger (ZnF) domain, and a nuclear localization signal (NLS) located close to its N-terminal disordered region. (B) AlphaFold3 model of PARP7 aa 230–657 with NAD (nicotinamide adenine dinucleotide) and $Zn^{2+}$ (Seed: 1707246408), ipTM: 0.95, pTm=0.87. (C) AlphaFold3 model of PARP7 aa 230-657 and TBA4A with NAD and $Zn^{2+}$ (Seed: 775869383), ipTM: 0.76, pTm=0.76. (D) AlphaFold3 model of PARP7 aa 230–657 and HIF1A aa 1–401 with NAD and $Zn^{2+}$ (Seed: 1166797532), ipTM: 0.65, pTm = 0.74. (E) AlphaFold3 model of PARP7 aa 230–657 and RelA aa 19–189 with NAD and $Zn^{2+}$ (Seed: 532216561), ipTM: 0.66, pTm = 0.78.

**Table 1. Known substrates of PARP7, their modification sites, and functional outcomes of MARylation.**

| Substrate | MARylation site(s) | Effects of MARylation | Effects of MARylation loss | Reference |
|---|---|---|---|---|
| PARP7 | C39, C100, C439, C543, C552 | Automodification, degradation, and formation of nuclear bodies | Protein stabilization, diffuse nuclear localization | Gomez et al, 2018; Rodriguez et al, 2021; Yang et al, 2021 |
| AHR | Within the following peptides: C493-E512, S588-E615, Q629-K641, L678-E690, Q743-Y765, Q773-M786, T810-E835 | Degradation, negative regulation | Active AHR signaling | Gomez et al, 2018; Lu et al, 2019; MacPherson et al, 2014 |
| AR (agonist-bound) | C125, C131, C284, C290, C327, C406, C519, C596, C602, C620, C670 | Can be MARylated only when agonist-bound, assembly of the DTX3L-PARP9 heterodimer | Inability to interact with the DTX3L-PARP9 heterodimer, loss of transcriptional regulation by DTX3L-PARP9 | Kamata et al, 2021a; Kamata et al, 2021b; Wijngaarden et al, 2023; Yang et al, 2023; Yang et al, 2021 |
| α-tubulin | E55, D69, E71, E77, E90, E290, E297 | Microtubuli instability, cancer motility, and proliferation | Increased microtubuli stability, reduced migration and proliferation | Palavalli Parsons et al, 2021 |
| c-Myc | Not known | Degradation, negative regulation | Active c-Myc signaling | Zhang et al, 2020 |
| ER | Within the following peptides: E61-Y73, L121-Y139, E143-R158 | Degradation, negative regulation | Active ER signaling | Rasmussen et al, 2023; Rasmussen et al, 2021; Zhang et al, 2020 |
| FRA1 | C97 | Protection from proteolytic degradation, protein stabilization, apoptosis inhibition, and cancer cell growth promotion | Proteolytic degradation, induction of immune signaling, and apoptosis genes | Manetsch et al, 2023; Naulin et al, 2024 |
| HIF-1α | Not known | Degradation, negative regulation, suppression of the Warburg effect | Active HIF-1α signaling | Zhang et al, 2020 |
| LXRα and LXRβ | Within the central and/or C-terminal regions | Positive regulation of LXR activity | Negative regulation of LXR activity | Bindesbøll et al, 2016 |
| PARP13.1 (13.2*) | C15*, C106*, C168*, C174*, C187*, C219*, C721 | Not known | Not known | Rodriguez et al, 2021 |
| RelA | Not known | Decreased protein levels of RelA | Increased protein levels of RelA, functional NF-κB signaling | Rasmussen et al, 2023 |
| TBK1 | Within the kinase domain | Impaired autophosphorylation and kinase activity, suppression of IFN-I signaling | Active signaling downstream of TBK1, induction of IFN-I | Gozgit et al, 2021; Yamada et al, 2016 |

MARylation sites with asterisks are present in PARP13.2 as well as PARP13.1, and the one without is present only in PARP13.1.

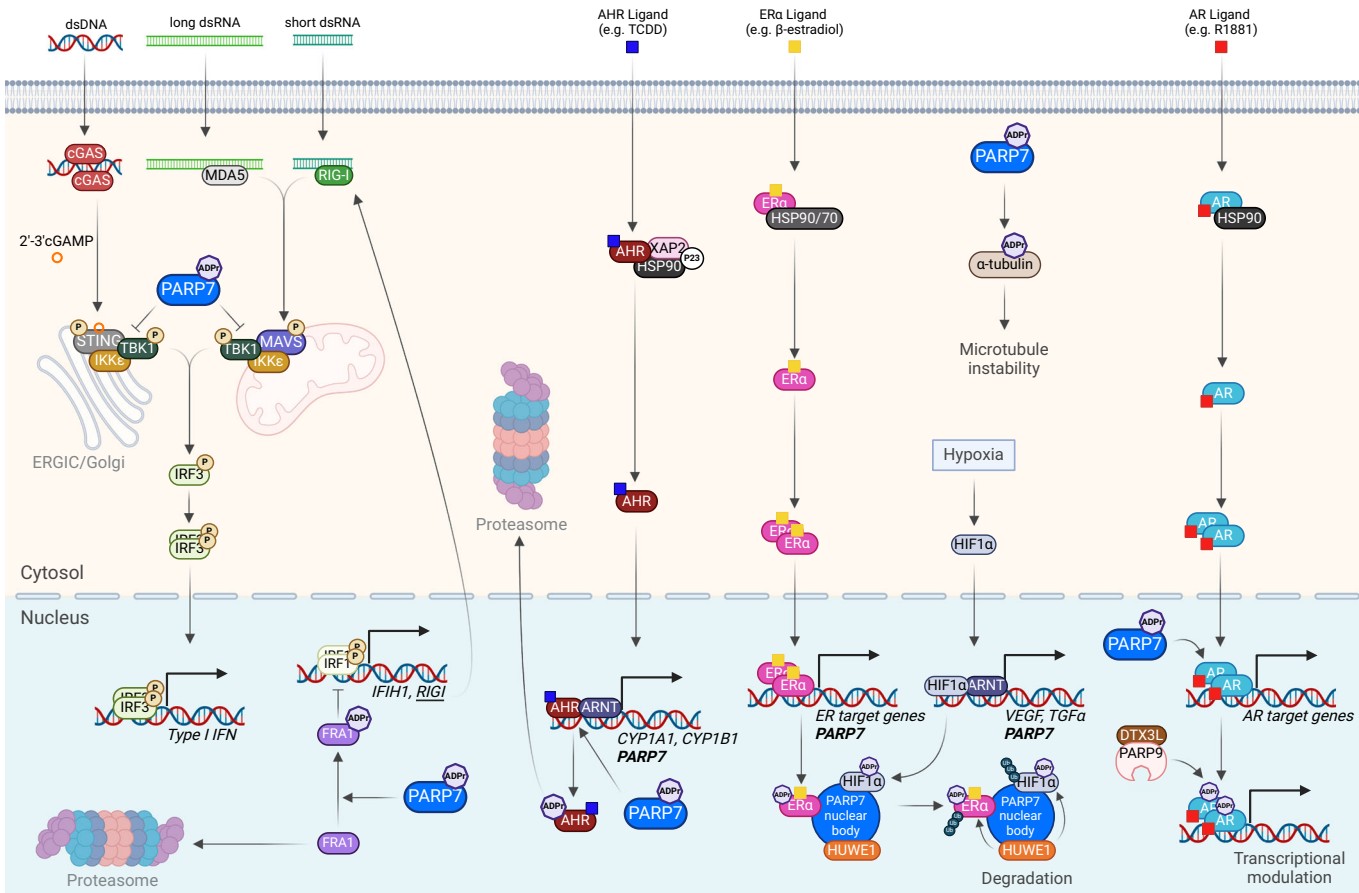

**Figure 2. Biological functions of PARP7.**

PARP7 is involved in type I interferon (IFN-I), aryl hydrocarbon (AHR), estrogen receptor (ER), and androgen receptor (AR) signaling. In mouse embryonic fibroblasts (but not mouse mammary cancer cells) it was shown that PARP7 ADP-ribosylates TBK1, preventing its autophosphorylation and thus its kinase activity, which impairs the IFN-I response. Cytosolic nucleic acids are recognized by nucleic acid sensors – cGAS for DNA and RIG-I/MDA5 for RNA. These then signal to STING or MAVS, respectively, both of which recruit TBK1 and IKKε, which phosphorylate and activate IRF3, resulting in IFN-I signaling. PARP7-mediated ADP-ribosylation stabilizes the transcription factor Fos-related antigen 1 (FRA1) and protects it from proteasomal degradation. ADP-ribosylated FRA1 negatively regulates IRF1 and thereby impairs RLR/MAVS-mediated activation of IRF3. PARP7 serves as a negative regulator of AHR signaling. Upon AHR-ligand binding, AHR is released and translocates to the nucleus, where it upregulates its target genes, one of which is PARP7 itself. PARP7 ADP-ribosylates AHR, thus marking it for degradation by the proteasome. Within the nucleus, PARP7 aggregates and forms nuclear bodies, which recruit and ADP-ribosylate transcription factors such as estrogen receptor α (ERα) and hypoxia-inducible factor 1α (HIF-1α). Those are then ubiquitinated with the help of E3 ubiquitin ligases, such as HUWE1, and degraded. In the context of androgen signaling, PARP7 ADP-ribosylates the ligand-bound androgen receptor (AR). The ADP-ribose moieties are then "read" by a heterodimer comprised of Deltex E3 ubiquitin ligase 3L (DTX3L) and PARP9, thereby modulating the transcription of a subset of AR-target genes. PARP7-mediated ADP-ribosylation reduces the stability of α-tubulin.

upregulation of their expression. AHR also directly regulates a long non-coding RNA (lncRNA) upstream of PARP7–TIPARP antisense RNA 1 (TIPARP-AS1) (Grimaldi et al, 2018). Reciprocally, PARP7 regulates AHR by MARylating key peptides within its transactivation domain, which leads to its proteolytic degradation and downregulation of AHR target genes (Ahmed et al, 2015; Gomez et al, 2018; Lu et al, 2019; MacPherson et al, 2014) (Table 1). TIPARP-AS1 reinforces the repressive effects of PARP7 (Grimaldi et al, 2018).

Later on, PARP7 emerged as a target gene and regulator of many other nuclear transcription factors, such as estrogen receptor (ER) (Rasmussen et al, 2021), androgen receptor (AR) (Kamata et al, 2021a; Kamata et al, 2021b), hypoxia-inducible factor 1α (HIF-1α) (Zhang et al, 2020), as well as the liver X receptors (LXRs) (Bindesbøll et al, 2016). PARP7 exerts its regulatory function by MARylating all of the aforementioned receptors and exhibits

context specificity, acting as a co-activator for some (e.g., LXRs) and as a co-repressor for others (e.g., AHR, ER, HIF-1α) (Bindesbøll et al, 2016; Kamata et al, 2021a; Kamata et al, 2021b; Rasmussen et al, 2021; Zhang et al, 2020) (Table 1; Figs. 1D and 2). As a co-activator of LXR, PARP7 positively regulates sterol regulatory element-binding protein 1 (SREBP1) and SREBP2, which are responsible for cholesterol and fatty acid import and synthesis (Bindesbøll et al, 2016).

PARP7 was shown to promote or suppress the toxic effects of AHR agonists. The AHR agonist 2,3,7,8-tetrachlorodibenzo-p-dioxin (TCDD) induces the lethal wasting syndrome characterized by the suppression of gluconeogenesis (Diani-Moore et al, 2010). PARP7 plays a major role in mediating the effects of TCDD in chick embryo hepatocytes, as it reduces the expression of phosphoenolpyruvate carboxykinase (PEPCK) and glucose-6-

phosphatase (G6Pase)—key enzymes of gluconeogenesis. By depleting NAD$^+$ levels in the liver, PARP7 reduces the activity of another NAD$^+$-dependent enzyme—sirtuin 1 (SIRT1), which is necessary for the activation of the gluconeogenic gene co-activator peroxisome proliferator-activated receptor γ co-activator 1-α (PGC1α). Conversely, in knock-out (KO) mouse models, PARP7 was shown to have protective effects by (i) suppressing TCDD-induced hepatosteatosis (fatty liver disease) (Ahmed et al, 2015), and (ii) suppressing the effects of the AHR agonist 3-methylcholanthrene (3MC) on chylous ascites (peritoneal fluid that is rich in triglycerides) (Cho et al, 2019).

## PARP7 in IFN-I signaling

PARP7 acts as a negative regulator of the IFN-I signaling, which mediates the innate immune response against viral infections or tumors. PARP7 can negatively regulate the IFN-I response by modulating AHR signaling, cGAS-STING, and RIG-I-MAVS pathways. A study by Yamada et al first revealed the important role of PARP7 as a negative regulator of the IFN-I response (Yamada et al, 2016). Activation of AHR signaling with tryptophan metabolites such as kynurenine (Kyn), an endogenous AHR ligand which is upregulated during inflammation and/or tumor progression, was shown to suppress the IFN-I response after infection with either RNA or DNA viruses (Yamada et al, 2016). Upon knockdown of PARP7 the suppressive effect of Kyn was abolished, indicating that PARP7 promotes the immunosuppressive effects of this AHR agonist. Some viruses such as mouse hepatitis virus (MHV) activate the AHR signaling, thereby upregulating PARP7 and suppressing the IFN-I response (Grunewald et al, 2020).

The IFN-I response relies on nucleic acid sensors in the cytosol: cGAS-STING and RIG-I-MAVS. The cGAS-STING pathway recognizes cytosolic DNA, whereas the RIG-I-MAVS and the MDA5-MAVS pathways are responsible for cytosolic RNA recognition. These distinct pathways converge on licensing TANK-binding kinase 1 (TBK1) to autophosphorylate itself, thus leading to interferon regulatory factor 3 (IRF3) phosphorylation, dimerization, and nuclear translocation, and the subsequent upregulation of *IFNB1*. In addition, TBK1 also contributes to the canonical NF-κB pathway by phosphorylating the IκB kinase complex (IKK), leading to the release of the RelA/p50 heterodimer and its translocation to the nucleus, where it upregulates *IFNB1* and inflammatory cytokines. Secreted IFNB1 binds to the interferon-α/β receptor (IFNAR) and activates JAK/STAT signaling, culminating in the expression of interferon-stimulated genes (ISGs) such as cytokines and chemokines that attract immune cells (Ablasser and Hur, 2020; Kwon and Bakhoum, 2020).

PARP7 was suggested to suppress the IFN-I response by ADP-ribosylating TBK1—the key kinase in cytosolic nucleic acid-sensing pathways. TBK1 ADP-ribosylation prevents its activation by autophosphorylation and thereby prevents IFN-I induction in MEFs (Yamada et al, 2016) (Fig. 2). In PARP7 KO MEFs infected with the vesicular stomatitis virus (VSV), pTBK1 levels were found to be increased (Yamada et al, 2016). This effect is dependent on the activation of the IFN-I signaling pathway as pTBK1 levels do not differ between PARP7 KO and WT untreated cells (Sanderson et al, 2023; Yamada et al, 2016). Moreover, pTBK1 levels were not increased in PARP7 KO EO771 mouse mammary cancer cells upon induction of the IFN-I signaling with the STING agonist

dimethylxanthenone acetic acid (DMXAA) and PARP7 inhibition did not increase pTBK1 levels in wild-type cells treated with DMXAA (Rasmussen et al, 2023). PARP7-mediated effects on pTBK1 may therefore be cell line- and treatment-dependent.

PARP7 likely has multiple substrates within the IFN-I pathway, as *IFNB1* induction was weaker in PARP7 KO cells compared to TBK1-depleted cells (Rasmussen et al, 2023). The RelA (p65) subunit of NF-κB was proposed as another substrate of PARP7 that could contribute to PARP7-mediated IFN-I response (Rasmussen et al, 2023) (Fig. 1E). Fos-related antigen 1 (FRA1) is a recently identified PARP7 substrate, which gets MARylated on C97 and is thus protected from proteasomal degradation (Manetsch et al, 2023; Naulin et al, 2024) (Table 1; Fig. 2). Stabilized FRA1 negatively regulates interferon regulatory factor 1 (IRF1), which is required for the induction of RIG-I-like receptor (RLR)-encoding genes, and thereby negatively regulates the RIG-I-MAVS-IRF3 signaling axis in lung cancer cells and suppresses caspase 8-driven apoptosis (Manetsch et al, 2023; Naulin et al, 2024).

Other than promoting the anti-viral innate immune response, IFN-I signaling is known to suppress intestinal inflammation by reducing the production of pro-inflammatory cytokines (Cho and Kelsall, 2014). Accordingly, in a murine model of dextran sulfate sodium (DSS)-induced colitis (intestinal inflammation), PARP7 loss was shown to reduce the expression of several inflammation genes (e.g., *IL6*, *Cxcl1*) and thereby reduce intestinal inflammation (Hutin et al, 2022).

## PARP7 in cancer

Given the established importance of PARP7 in the innate immune response and many signaling pathways that affect cell proliferation and metabolism, its deregulation is expected to contribute to various diseases including cancer.

PARP7 is ubiquitously expressed, with the highest expression levels in the skin, central nervous system (CNS), and reproductive organs, and lowest levels in the pancreas, kidney, and liver (Fig. 3A). PARP7 is most frequently mutated in endometrial carcinoma, followed by melanoma (Fig. 3B). Missense point mutations are the most prevalent type of mutations, the most common one being C257Y within the CCCH domain (Fig. 3C,D). PARP7 is downregulated in breast cancer, liver hepatocellular carcinoma, lung adenocarcinoma, lung squamous cell carcinoma, and stomach adenocarcinoma (Fig. 3E). PARP7 is overexpressed in pancreatic cancer (PAAD) and lower expression levels of PARP7 correlate with a better outcome in this cancer type (Figs. 3E and 4). Lower PARP7 expression levels also correlate with a better outcome in cervical squamous cell carcinoma (CESC), sarcoma (SARC), liver hepatocellular carcinoma (LIHC), lung squamous cell carcinoma (LUSC), lung adenocarcinoma (LUAD), head and neck squamous cell carcinoma (HNSC), and bladder carcinoma (BLCA), but with a worse outcome in uveal melanoma (UVM), rectum adenocarcinoma (READ), kidney renal clear cell carcinoma (KIRC), breast cancer (BRCA), thyroid cancer (THCA), and prostate adenocarcinoma (PRAD) (Fig. 4).

In the ER-positive human breast cancer cells MCF7, siRNA-mediated depletion or inhibition of PARP7 using RBN-2397 promotes their proliferation in vitro and in vivo (Zhang et al, 2020; Rasmussen et al, 2021) (Table 2). β-estradiol induces the expression of

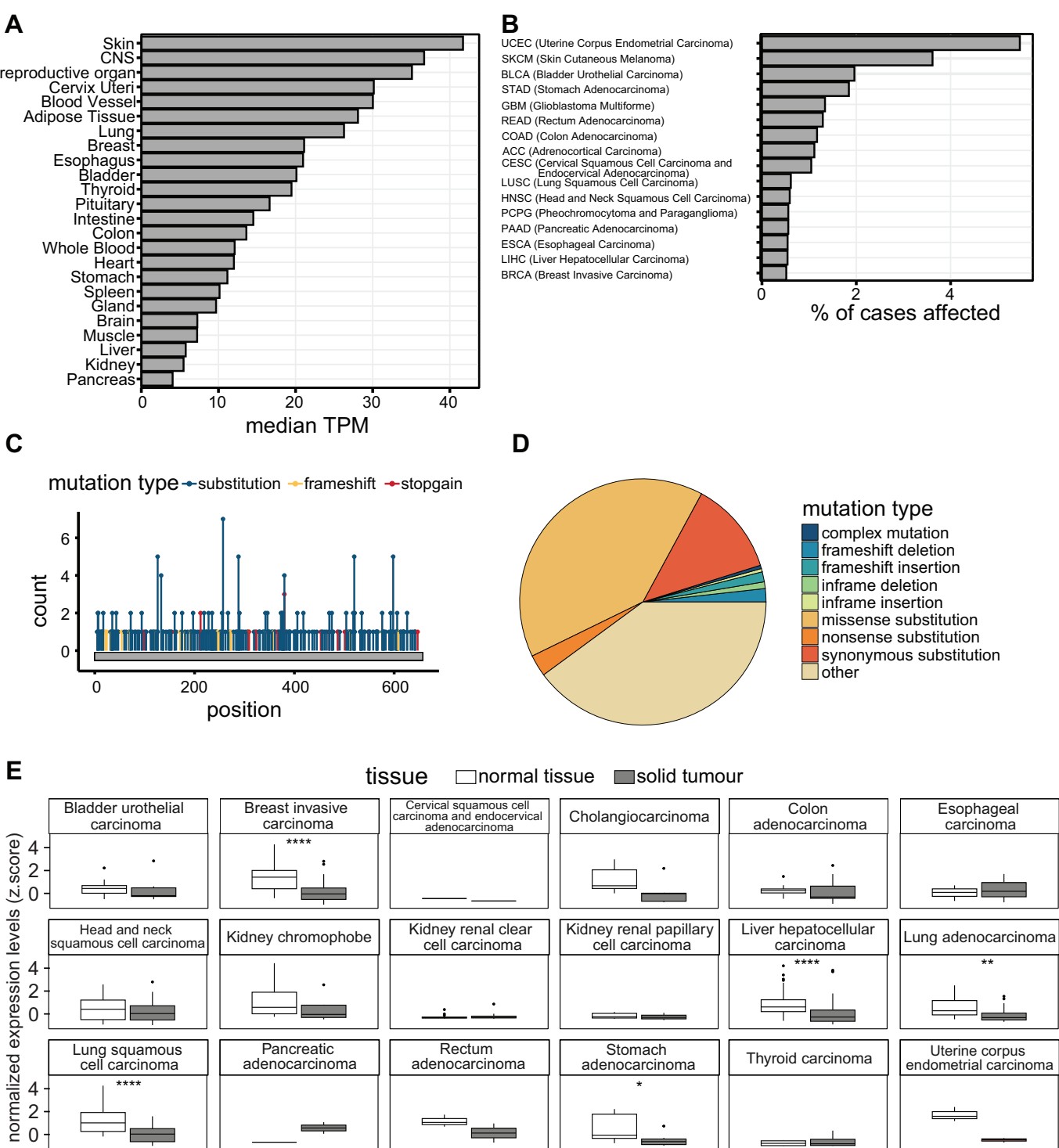

**Figure 3. PARP7 expression levels and cancer-associated mutations.**

(**A**) PARP7 (TIPARP) gene expression levels [transcripts per million (TPM)] per tissue accessed from the GTEx portal (http://www.gtexportal.org). (**B**) The top ten TCGA cohorts with PARP7 (TIPARP) mutation ranked by the percentage of affected cases [accessed via the GDC (Genomic Data Commons) data portal (https://portal.gdc.cancer.gov)]. (**C**) Type and frequency of somatic mutations of PARP7 (TIPARP) per residue. (**D**) Frequency distribution of the different PARP7 (TIPARP) mutation types. Data in (**C**, **D**) were accessed via the COSMIC (Catalogue of Somatic Mutations in Cancer) data portal (https://cancer.sanger.ac.uk). (**E**) PARP7 (TIPARP) normalized expression levels (z.score) of paired tumor–normal tissue across TCGA cohorts accessed via the R client FirebrowseR (Deng et al, 2017).

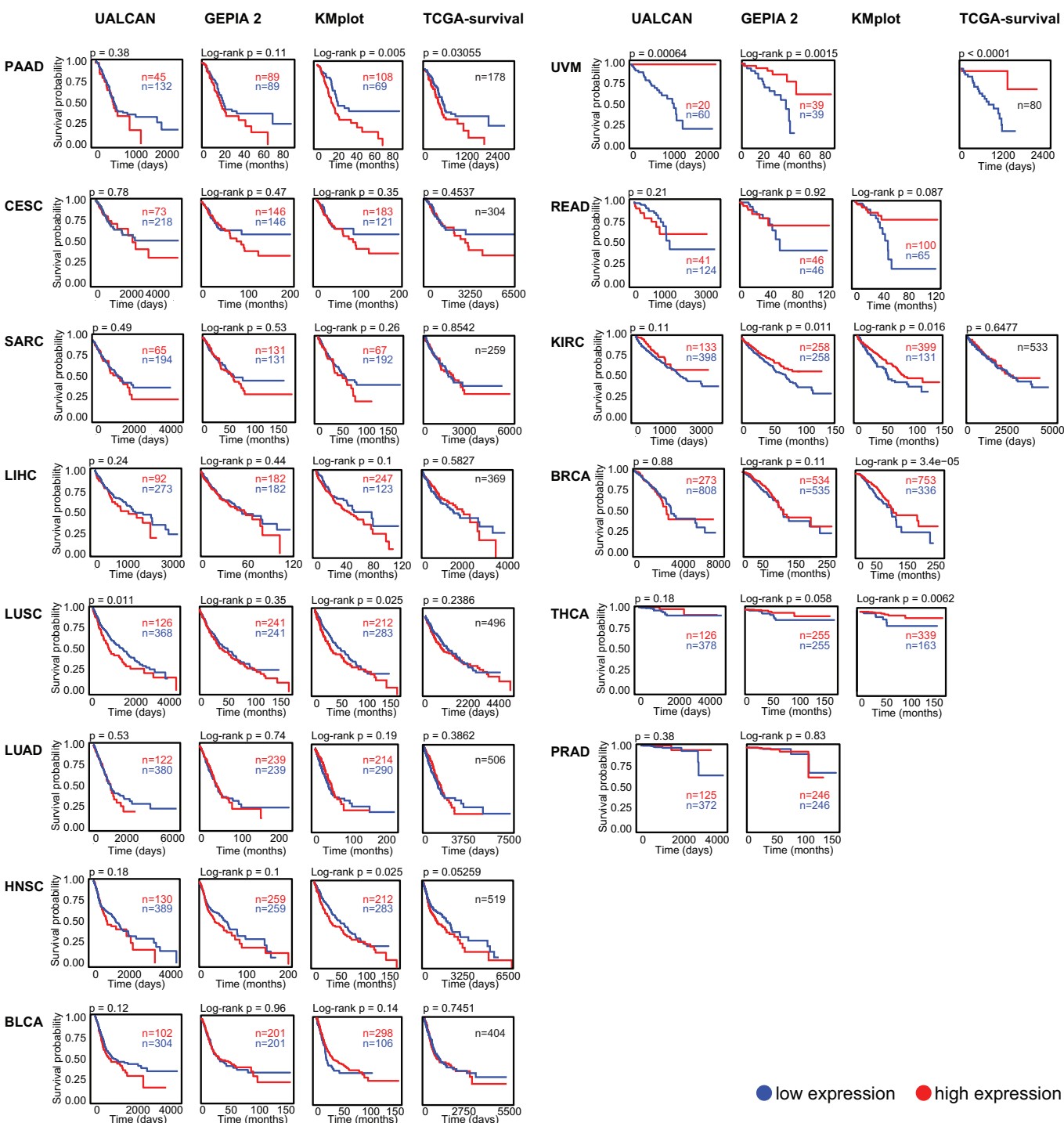

**Figure 4. Patient survival stratified based on PARP7 (TIPARP) expression levels measured by RNA-sequencing.**

Kaplan–Meier plots were accessed via the data portals UACLAN (Chandrashekar et al, 2017; Chandrashekar et al, 2022), GEPIA 2 (Tang et al, 2019), KMplot (https://www.kmplot.com) (Győrffy, 2024a, b), as well as TCGA-survival (https://www.tcga-survival.com) (Smith and Sheltzer, 2022). PAAD pancreatic adenocarcinoma, CESC cervical squamous cell carcinoma, SARC sarcoma, LIHC liver hepatocellular carcinoma, LUSC lung squamous cell carcinoma, LUAD lung adenocarcinoma, HNSC head and neck squamous cell carcinoma, BLCA bladder carcinoma, UVM uveal melanoma, READ rectum adenocarcinoma, KIRC kidney renal clear cell carcinoma, BRCA breast cancer, THCA thyroid cancer, PRAD prostate adenocarcinoma.

**Table 2. PARP7 functions in cancer, relevant pathways, and outcome of PARP7 depletion, inhibition or overexpression.**

| Cancer type | | PARP7 tumor effects | PARP7 substrates/pathways involved | PARP7 depletion/inhibition outcome | Reference |
|---|---|---|---|---|---|
| Breast cancer | MCF7 (xenograft model) | Negative regulation of ERα | ERα signaling | Depletion or inhibition of PARP7 via RBN-2397 promotes proliferation and tumor growth by stabilizing ERα and its signaling | Zhang et al, 2020; Rasmussen et al, 2021 |
| | EO771 (syngeneic model; Parp7$^{H532A}$) | Tumor growth; reduced levels of IFNB1 | TBK1- and NF-κB-dependent IFN-I signaling | PARP7 loss, inhibition, or catalytic mutant lead to reduced tumor growth and anti-tumor immunogenic effects via increased IFN-I signaling and immune cell recruitment | Rasmussen et al, 2023 |
| Ovarian cancer | OVCAR4 and OVCAR3 | Tumor growth, proliferation, and motility | α-tubulin | PARP7 depletion or inhibition stabilizes microtubules, thus reducing cancer migration and invasion, and upregulates cell-cell adhesion, apoptosis, and cell-cycle arrest genes | Palavalli Parsons et al, 2021; Spirtos et al, 2024 |
| Cervical cancer | HeLa | | | | |
| Kidney cancer | A704 | | | | |
| Lung cancer | A549 (overexpression study) | Angiogenesis and remodeling of the extracellular matrix | Not known | No inhibition/depletion data | Miura et al, 2023 |
| | Non-small cell lung cancer (NCI-H1373 and NCI-H1975) | Proliferation, negative regulation of IFN-I signaling | FRA1, IRF1, RLR-MAVS-mediated IRF3 activation | PARP7 inhibition promotes FRA1 degradation and thus induces the expression of apoptosis and immune signaling genes | Manetsch et al, 2023 |
| Prostate cancer | VCaP, PC3, PC3M | MARylation of AR allows for gene expression modulation by the DTX3L-PARP9 complex | AR signaling | PARP7 depletion causes dysregulation of AR-dependent transcription | Kamata et al, 2021a; Kamata et al, 2021b; Yang et al, 2021 |

PARP7 via ER (Zhang et al, 2020) and PARP7 in turn has a negative effect on β-estradiol-mediated proliferation by MARylation of at least three motifs located in the N-terminal AF-1 domain of ER, marking it for degradation (Rasmussen et al, 2021) (Table 1; Fig. 3). PARP7 inhibition led to stabilization of ER and therefore enhanced the expression of the ER signaling target gene *GREB1*, contributing to increased proliferation.

In the ER-positive murine breast cancer cell line EO771, PARP7 had no effect on proliferation but was found to dampen TBK1- and NF-κB-dependent IFN-I signaling, while PARP7 inhibition was shown to stimulate IFN-I signaling and immune cell infiltration (Rasmussen et al, 2023) (Table 2). PARP7 KO EO771 cells injected in immunodeficient mice showed the same magnitude of tumor growth as WT cells. However, in immunocompetent PARP7$^{+/+}$ mice injection of PARP7 KO EO771 cells resulted in increased IFN-I signaling and higher T cell infiltration compared to WT cells. In PARP7 catalytic mutant (PARP7$^{H532A}$) mice injection of either WT or PARP7 KO EO771 cells led to a significantly increased infiltration of pro-inflammatory M1 macrophages and a higher reduction in tumor growth compared to PARP7 WT mice. Taken together, loss of PARP7 in either cancer cells or immune cells induces anti-tumor immunogenic effects. However, to circumvent the tumor-promoting effects of PARP7 inhibition found in MCF7 cells, it may be advantageous to administer it selectively to immune cells.

PARP7 promotes ovarian cancer growth and motility. PARP7 knockdown in OVCAR4 cells reduces their proliferation, as well as their migration and invasion (Palavalli Parsons et al, 2021) (Table 2). RNA-seq data analysis further revealed that depletion of PARP7 in OVCAR4 cells leads to the upregulation of genes associated with cell-cell adhesion, apoptosis, and cell-cycle arrest. Moreover, the knockdown of PARP7 promotes microtubule stability not just in OVCAR4 cells but also in OVCAR3, HeLa, and A704 (kidney adenocarcinoma) cells through α-tubulin as a direct target of PARP7-mediated MARylation (Palavalli Parsons et al, 2021) (Table 1; Figs. 1C and 2). In a study by Spirtos et al, a synergistic effect was observed between the PARP7 inhibitor RBN-2397 and paclitaxel on the microtubule network in ovarian cancer (Spirtos et al, 2024). While RBN-2397 leads to the stabilization of microtubules by preventing MARylation of α-tubulin, paclitaxel stabilizes β-tubulin and prevents microtubule depolymerization. When used in combination, both drugs enhance each other's therapeutic efficacy and significantly reduce ovarian cancer motility (Spirtos et al, 2024).

In the lung cancer cell line A549, overexpression of PARP7 was shown to promote angiogenesis upon engraftment onto the chorioallantoic membrane of chick embryos (Miura et al, 2023) (Table 2). Genes relevant to the extracellular matrix (ECM) and its remodeling were found to be upregulated, whereas genes involved in pro-inflammatory pathways were downregulated. Elevated expression of PARP7 in human lung cancer could induce angiogenesis and thus promote tumor growth and metastasis by altering the ECM and suppressing the production of pro-inflammatory cytokines.

The non-small cell lung cancer (NSCLC) cell lines NCI-H1373 and NCI-H1975 were shown to be highly dependent on PARP7 for growth and sensitive to PARP7 inhibition or knockout (Gozgit et al, 2021; Manetsch et al, 2023; Naulin et al, 2024) (Table 2). The sensitivity of NCI-H1975 cells to PARP7 inhibition was linked with the degradation of the transcription factor FRA1. FRA1 negatively

regulates IRF1 and impairs RLR-MAVS-mediated IRF3 activation and expression of genes related to apoptosis as well as immune signaling in NCI-H1975 cells (Manetsch et al, 2023). PARP7 stabilizes FRA1 by preventing its proteasomal degradation via MARylation (Manetsch et al, 2023). PARP7 inhibition can thus induce apoptosis and immune signaling by promoting FRA1 degradation.

PARP7 has a higher expression in tumor than in normal thyroid tissue and was identified as one of the eight key estrogen-related differentially expressed genes (ERDEGs) in thyroid cancer (THCA) (Zhang et al, 2023a). Increased levels of PARP7 are associated with male sex, advanced papillary thyroid carcinoma (PTC) stage, and lymph node metastasis (LNM); PARP7 may thus serve as a prognostic marker for LNM in male PTC (Zhang et al, 2024).

PARP7 levels are also increased in androgen-dependent prostate cancer. Androgen receptor (AR) signaling is implicated as the main driver mechanism of prostate cancer (PCa). Upon androgen binding, AR adopts an active (agonist) conformation, which allows for subsequent transcriptional regulation of its target genes. PARP7 is a direct target gene of AR, and its levels are stabilized by AR signaling (Kamata et al, 2021a). A distinct AR-PARP7 signaling axis has been elucidated, where PARP7 acts as an ADPr writer, and PARP9 in a complex with the histone E3 ubiquitin ligase DTX3L acts as a reader (Yang et al, 2021). Upon binding of R1881, a synthetic androgen agonist, AR undergoes a conformational change, which is required for the PARP7-mediated MARylation on key cysteine residues within its N-terminal domain (Kamata et al, 2021b) (Table 1; Fig. 2). Furthermore, androgen induces the association of AR with the DTX3L-PARP9 heterodimer, where PARP9, using its macrodomains (MD1 and MD2), binds to the ADP-ribose moieties on AR deposited by PARP7, thus modulating AR transcriptional activity both positively and negatively in prostate cancer cells (Yang et al, 2021). In addition, a key role for the dual MARylation of the cysteine pairs C125/C131 and C284/C290 of AR emerged, as the DTX3L-PARP9 complex exhibits higher affinity for this dual modification (Wijngaarden et al, 2023).

In addition to its pro-tumorigenic effects, PARP7 may also have anti-tumor effects by encapsulating multiple oncogenic transcription factors (HIF-1α, c-Myc, and ER) and facilitating their degradation via MARylation within condensates (Zhang et al, 2020). PARP7 promotes protein degradation by interacting with a series of E3 ubiquitin ligases, one of which is the WWE domain containing HUWE1 (Zhang et al, 2020). PARP7 forms nuclear condensates with HUWE1 facilitated by its own WWE domain, which recognizes and binds to ADP-ribosylated proteins, including PARP7 itself (Zhang et al, 2020) (Fig. 2).

# PARP7 inhibitors

The development of PARP7 inhibitors was prompted by the reported tumorigenic effects of PARP7 and the necessity to further unravel the catalytic functions of PARP7 in cells. The first specific PARP7 inhibitor was RBN-2397 (atamparib), which has reached clinical trials (Gozgit et al, 2021; Nizi et al, 2022) (Tables 3, 4; Fig. 5). In the following years, optimization of the PARP1/2/7 inhibitor—Phthal01, led to the development of the next PARP7-specific inhibitor—KMR-206, the scaffold of which differs from that of RBN-2397 (Rodriguez et al, 2021; Sanderson et al, 2023).

Both inhibitors increase endogenous PARP7 protein levels, suggesting that its catalytic activity regulates its protein levels (Sanderson et al, 2023). PARP7 protein levels are higher for the catalytically inactive mutant and upon proteasome inhibition (Lu et al, 2019), and the ubiquitin ligase HUWE1 only interacts with WT PARP7 (Zhang et al, 2020), suggesting that PARP7 stability is regulated by its catalytic activity in a ubiquitin-dependent manner. In addition, both inhibitors lead to the accumulation of endogenous PARP7 in the nucleus in a diffuse manner, encompassing the entire nucleus, as opposed to forming distinct foci in the absence of inhibitors (Sanderson et al, 2023).

## RBN-2397 and its derivatives

RBN-2397 contains a pyridazinone scaffold (Fig. 5) and inhibits PARP7 with an IC$_{50}$ of <3 nM [based on Time-Resolved Fluorescence Energy Transfer (TR-FRET)], 1 nM [based on Nanoluciferase Bioluminescence Resonance Energy Transfer (NanoBRET)] 0.22 nM [based on Surface Plasmon Resonance (SPR)] by competing with NAD$^+$ for the binding pocket (Table 3). Inhibition of PARP7 prevents MARylation and promotes phosphorylation of TBK1, resulting in the restoration of the IFN-I signaling in CT26 colon carcinoma cells (Gozgit et al, 2021). Although RBN-2397 shows clear selectivity for PARP7, the IC$_{50}$ values for PARP1, PARP2, and PARP12 are 37 nM, 17 nM, and 25 nM respectively [based on Dissociation-Enhanced Lanthanide Fluorescent Immunoassays (DELFIA)] (Gozgit et al, 2021), raising the question as to whether the observed effects are due to PARP7 inhibition alone. RBN-2397-mediated effect on the IFN-I response, as assessed by the induction of CXCL10, was comparable in wild-type and PARP1, PARP3 or PARP12 knock-out (KO) CT26 cells, but was reduced in PARP2 KO cells, in accordance with the smallest difference in IC$_{50}$ between PARP7 and PARP2 (Gozgit et al, 2021).

In CT26 tumor-bearing mice, RBN-2397 induced durable tumor regression, which is mediated by CD8+ T cells and enhanced in combination with anti-PD-1 antibodies (Gozgit et al, 2021). Its anti-tumor effects are dependent on the presence of PARP7, STING and TBK1 (Gozgit et al, 2021). Mechanistically, PARP7 inhibition through RBN-2397 induces pIRF3 and pSTAT1 in TBK1- and STING-dependent manner in CT26 cells. A CRISPR screen confirmed that anti-proliferation effects of RBN-2397 in NCI-H1373 cells rely on the IFN-I signaling, as depletion of STING, IRF3, IFNB1, STAT2, and IRF9 led to resistance to PARP7 inhibition (Gozgit et al, 2021). Accordingly, cancer cell lines that have increased expression of ISGs at baseline are more responsive to PARP7 inhibition. These promising pre-clinical results prompted clinical trials in solid tumors and NSCLC alone or in combination with anti-PD-1 antibody (pembrolizumab; Table 4).

Conversely, RBN-2397 exerted its growth-inhibitory effects in an IFN-independent manner in different PCa cell lines (Yang et al, 2023). Induction of PARP7 via AR or AHR agonists sensitized them to RBN-2397 inhibition. Moreover, RBN-2397 prevented AR MARylation and the formation of the AR-DTX3L-PARP9 complex, induced chromatin "trapping" of PARP7, and increased the levels of the cell cycle regulator cyclin-dependent kinase inhibitor (CDKN1A) in PCa cells.

In addition to PARP7 itself, several gene knock-outs conferred resistance to RBN-2397, including the known PARP7 substrate AHR. Conversely, the addition of AHR agonists such as L-kynurenine and tapinarof further sensitized the cells (Chen et al, 2022). Inhibition of

Table 3. PARP7-specific inhibitors, their potency, pharmacokinetic properties, and effects.

| | Inhibitor | IC$_{50}$ (nM) | | AUC (h*ng/mL) | Bioavailability (F) | TGI | Effects | Reference |
|---|---|---|---|---|---|---|---|---|
| RBN-2397 and its derivatives | RBN-2397 | PARP7 | <3 (TR-FRET); 1 (NanoBRET); 0.22 (SPR) | 656 | 25.67% | 49 – 67% (30% without anti-PD-1 antibody) | Inhibits catalytic activity, restores IFN-I signaling, increases infiltration of CD8+ T cells, traps PARP7 on chromatin, can induce toxicity at dose >100 mg/kg | Chen et al, 2022; Gozgit et al, 2021; Gu et al, 2023; Gu et al, 2024; Nizi et al, 2022; Sanderson et al, 2023; Xu et al, 2024; Yang et al, 2023; Zhang et al, 2023b |
| | | PARP1 | 37 ± 10 | | | | | |
| | | PARP2 | 17 ± 9 | | | | | |
| | | PARP12 | 25 ± 5 | | | | | |
| | I-1 | PARP7 | 7.6 ± 0.9 | 5663.54 | 95.80% | 67% | Inhibits catalytic activity, restores IFN-I signaling, increases infiltration of CD8+ T cells | Gu et al, 2023 |
| | | PARP1 | >5000 | | | | | |
| | | PARP2 | 546 ± 16 | | | | | |
| | | PARP12 | 453 ± 30 | | | | | |
| | (S)-XY-05 | PARP7 | 4.5 ± 0.01 | 16999 | 94.60% | 83% | Inhibits catalytic activity, restores IFN-I signaling, increases infiltration of CD8+ T cells | Gu et al, 2024 |
| | | PARP1 | >1300 | | | | | |
| | | PARP2 | 302.8 ± 1.3 | | | | | |
| | | PARP12 | 664 ± 32 | | | | | |
| | Compound 18 | PARP7 | 0.56 ± 0.02 | 1146 (in mice) 7860 (in Beagle dogs) | 33.9% (in mice) 45.2% (in Beagle dogs) | 75.2% | Inhibits catalytic activity, restores IFN-I signaling, inhibits cytochrome P450 enzymes, tolerable inhibition of the hERG potassium ion channel (dose <10 μM) | Zhang et al, 2023b |
| | | PARP1 | 492 ± 33 | | | | | |
| | | PARP2 | 221 ± 16 | | | | | |
| | | PARP12 | 360 ± 29 | | | | | |
| | Compound 8 | PARP7 | 0.11 ± 0.02 | 3430 (in mice) 1040 (in rats) 4410 (in Beagle dogs) | 104% (in mice) 30% (in rats) 78% (in Beagle dogs) | 81.6% | Inhibits catalytic activity, restores IFN-I signaling, weakly inhibits cytochrome P450 enzymes, inhibits the hERG potassium ion channel | Xu et al, 2024 |
| | | PARP1 | >1000 | | | | | |
| | | PARP2 | >1000 | | | | | |
| | | PARP12 | >1000 | | | | | |
| KMR-206 and its derivatives | KMR-206 | PARP7 | 13.7 | No in vivo data | | | Inhibits catalytic activity, restores IFN-I signaling to a lesser extent than RBN-2397 | (Sanderson et al, 2023) |
| | | PARP1 | >3000 | | | | | |
| | | PARP2 | 1015 | | | | | |
| | Phthtal01 | PARP7 | 14 ± 2 | No in vivo data | | | Inhibits catalytic activity | Lu et al, 2019; Rodriguez et al, 2021; Sanderson et al, 2023 |
| | | PARP1 | 21 ± 4 | | | | | |
| | | PARP2 | 28 ± 5 | | | | | |
| Pan-PARP inhibitors | Thioparib | PARP7 | 149 | No data | | 61.26% – 99% | Inhibits PARP1, represses HR DNA repair, induces IFN-I signaling via PARP7-STING/TBK1 signaling, overcomes olaparib resistance | Wang et al, 2023 |
| | | PARP1 | 0.4 | | | | | |
| | | PARP2 | 0.3 | | | | | |
| | Cpd36 | PARP7 | 0.21 | 617 | 14.7% | 54.6 – 94.3% | Inhibits catalytic activity, increases infiltration of CD8+ T cells, traps PARP1 onto DNA | Zhou et al, 2023 |
| | | PARP1 | 0.94 | | | | | |
| | | PARP2 | 0.87 | | | | | |
| Bifunctional conjugates | B3 | PARP7 | 2.5 ± 0.4 | 1524 | 69.1% | 28.20 – 40.43% | Inhibits both PARP7 and PD-1/PD-L1, restores T cell function, increases IFN-γ secretion | Gao et al, 2024 |
| | | PD-1/PD-L1 | 426 ± 88 | | | | | |
| | | PARP1 | 111.8 ± 13.6 | | | | | |
| | | PARP2 | 11.0 ± 0.9 | | | | | |
| | C6 | PARP7 | 7.1 ± 2.3 | 3022 | 50.6% | 21.12 – 40.04% | | |
| | | PD-1/PD-L1 | 343 ± 52 | | | | | |
| | | PARP1 | >5000 | | | | | |
| | | PARP2 | 5.6 ± 1.4 | | | | | |

The half-maximal inhibitory concentration (IC$_{50}$) is given in nanomolar (nM). The area under the curve (AUC) is given in hours (h) x nanograms (ng) per milliliter (mL). Tumor growth inhibition (TGI) is given in percentage. Values for PARP7 are indicated in bold letters.

**Table 4. Current clinical trials with PARP7 inhibitors either as a mono or combinatorial therapy.**

| Clinical trial number | Treatment | Organization | Type of tumor | Stage of trial |
|---|---|---|---|---|
| NCT04053673 | RBN-2397 | Ribon Therapeutics, Inc. | Solid tumors | Unknown status |
| NCT05127590 | RBN-2397 + Pembrolizumab | Ribon Therapeutics, Inc. | Squamous Non-Small Cell Lung Carcinoma | Active, not recruiting |
| NCT06433726 | BY101921 | Chengdu Baiyu Pharmaceutical Co., Ltd. | Solid tumors | Recruiting |

Data acquired from http://clinicaltrials.gov.

PARP7 leads to accumulation of AHR in the nucleus, suggesting that some of the anti-proliferative properties of RBN-2397 may be attributed to the active AHR signaling, which could end in AHR-dependent apoptosis due to upregulation of CYP1A1 and CYP1B1 (Chen et al, 2022). Another CRISPR screen in the same cell line revealed the cohesin complex as an additional determinant of resistance to PARP7 inhibition (Chen et al, 2022). PARP7 inhibition reduces SMC1A ADP-ribosylation, suggesting that SMC1A is a PARP7 substrate (Gozgit et al, 2021).

Anti-tumor immunogenic effects of RBN-2397 can be enhanced in PARP2, 3, 12, and 1/2 KO CT26 tumors while anti-proliferation effects are enhanced in PARP1 and PARP3 KO NCI-H1373 cells, suggesting that combinatorial treatments of different PARP inhibitors may sensitize colon and lung tumors. Sensitization was also observed in a CRISPR screen with knockouts of ADAR, DDX21, PRKDC (DNA-PK), EIF4A3, ATRX, and DHX9 as potential substrates of PARP7 that show loss of ADP-ribosylation upon PARP7 inhibition (Gozgit et al, 2021).

Given the success of RBN-2397, Gu et al developed an optimized inhibitor via cyclization of the ethyoxyl linkage (Gu et al, 2023) (Fig. 5B). Cyclization, in the context of inhibitors, can enhance specificity and potency by introducing a ring structure in the scaffold of the compound, thus conferring rigidity. Replacing the ethyoxyl linkage of the original molecule with an azetidine fragment led to the discovery of a highly potent ($IC_{50}$ = 7.6 nM) and PARP7-specific compound—I-1. Furthermore, I-1 demonstrated superior pharmacokinetic properties, such as lower clearance and high absorption when compared to RBN-2397. In vivo studies of CT26 tumor-bearing immunocompetent mice revealed a stronger inhibitory activity of I-1 on tumor growth compared to RBN-2397 at the same dose. Additionally, *IFNB1* and *CXCL10* mRNA levels and T-cell infiltration were higher in CT26 tumor tissues treated with I-1. In a later study, by further reducing the flexibility of RBN-2397, Gu et al synthesized indazole-7-carboxamide derivatives, with *(S)*-XY-05 exhibiting excellent selectivity and potency for PARP7 ($IC_{50}$ = 4.5 nM) (Gu et al, 2024). Not only was the oral bioavailability higher for *(S)*-XY-05 than for RBN-2397 (94.60% and 25.67%, respectively), but in vivo studies also demonstrated its remarkable tumor growth inhibition, with moderate effects observable even at doses as low as 6.25 mg/kg. *(S)*-XY-05 dose-dependently increased *IFNB1* mRNA levels in CT26 tumor tissues, whereas 100 mg/kg RBN-2397 failed to achieve a similar effect.

Exploring cyclization further as a prospective avenue for enhancing the selectivity of RBN-2397 towards PARP7 led to the development of compound 18 (Zhang et al, 2023b) and compound 8 (Xu et al, 2024). Using RBN-2397 as a starting point, Zhang et al generated a series of tricyclic derivatives, of which compound 18 exhibited an exceptionally high potency

against PARP7 ($IC_{50}$ = 0.56 nM), a longer half-life, a higher plasma concentration, and a higher tumor growth inhibition in NCI-H1373 tumor-bearing immunodeficient mice than RBN-2397 (Zhang et al, 2023b). However, evaluation of potential drug toxicities demonstrated that compound 18 inhibited enzymes of the cytochrome P450 family, which are responsible for the proper metabolization of a variety of drugs, as well as exhibited some weak cardiac toxicity. In a similar study by Xu et al another tricyclic compound – compound 8, was obtained (Xu et al, 2024). Crystal structure analysis revealed that the key differences between the adenosine subpocket of PARP7 and PARP2 is what allows for cyclization to be a valid strategy of conferring further selectivity for PARP7. Compound 8 not only inhibited PARP7 with an $IC_{50}$ of only 0.11 nM and exerted anti-proliferative effects in NCI-H1373 cells at $IC_{50}$ of 2.5 nM, but was also highly selective for it over other ART family members. Moreover, compound 8 had superior oral bioavailability in mice (104%) and Beagle dogs (78%) compared to RBN-2397 (25.67% in mice). In vivo studies in mice demonstrated that compound 8 inhibited NCI-H1373 tumor growth with a much higher potency than RBN-2397 at the same dose. Despite promising inhibitory properties and selectivity profile, concentrations higher than 10 µM of compound 8 exerted prominent inhibition of the hERG potassium ion channel, indicating high cardiac toxicity. Furthermore, compound 8 exhibited varying degrees of inhibition of a series of drug-metabolizing enzymes belonging to the cytochrome P450 family.

## KMR-206 and its derivatives

Rodriguez et al synthesized an analog of AZ126299495—a phthalazinone-based piperazine PARP1/2/7 inhibitor with nanomolar potency named Phthal01 (Lu et al, 2019; Rodriguez et al, 2021) (Fig. 5, Table 3). Phthal01 treatment showed loss of PARP7 auto-MARylation and trans-MARylation of both the full-length and truncated isoforms of PARP13 (PARP13.1 and PARP13.2, respectively) in HEK293T cells. Despite Phthal01 being 13-fold more selective for PARP7, it still exhibited double-digit nanomolar potency against PARP1 and PARP2 and at higher concentrations (~3 µM) it inhibited PARP10 auto-MARylation. The selectivity for PARP7 was further increased by Sanderson et al, who identified a hydrophobic cavity near the NAD+ binding site of MARylating ARTs, a trait not present in PARylating ARTs due to their conserved glutamate at the gatekeeper position (Sanderson et al, 2023). The addition of a propynyl group at the C-6 position of the Phthal01 scaffold fits inside the hydrophobic pocket, thus increasing the specificity of the new compound termed KMR-206 for PARP7 over PARP1 and PARP2, which also shows high nanomolar potency against PARP7 ($IC_{50}$ = 13.7 nM). Since RBN-2397 and KMR-206 are structurally distinct, treatment of

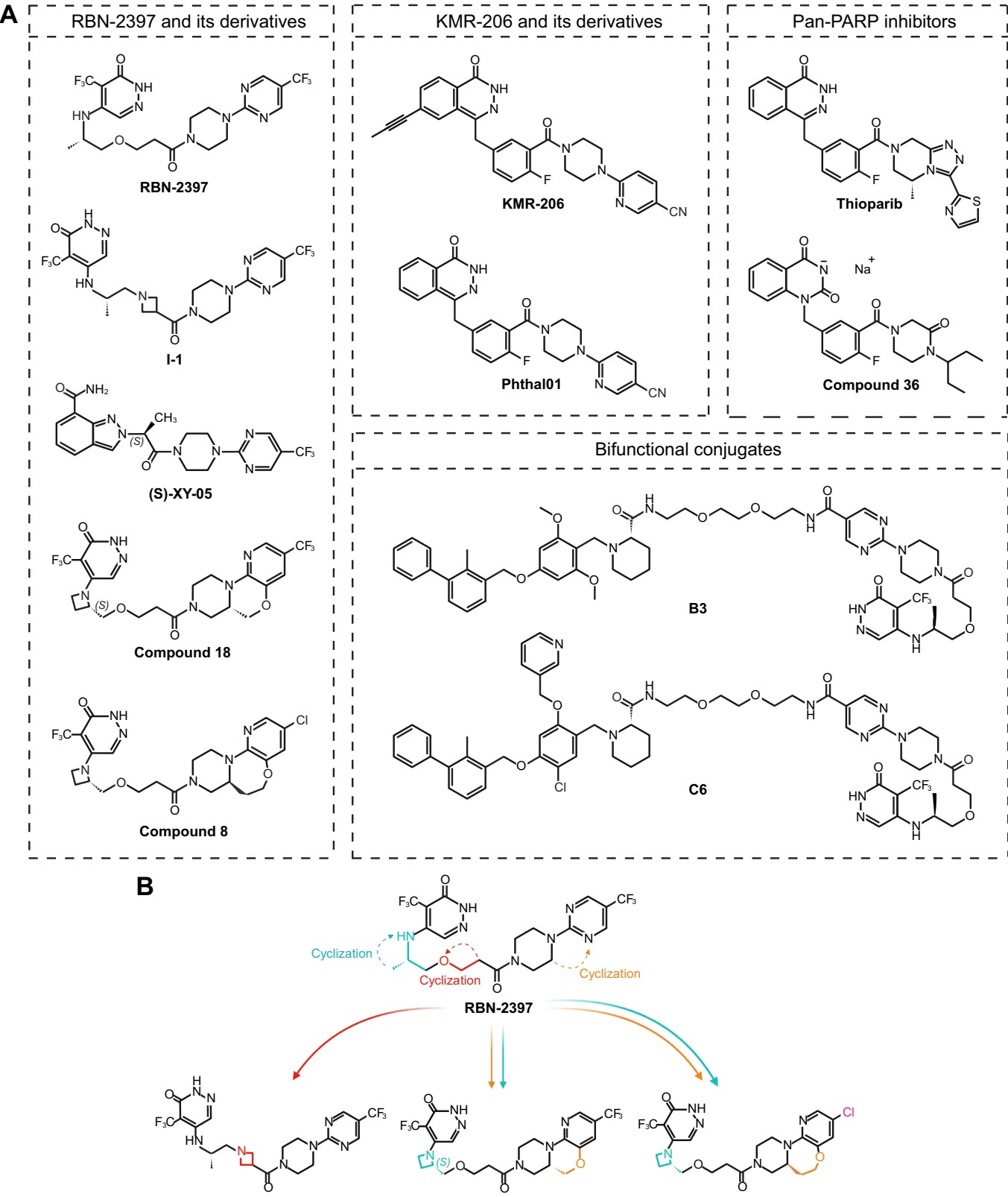

**Figure 5.  PARP7 inhibitor structures.**

(A) Chemical structures of various PARP7-targeting inhibitors. RBN-2397 comprises an NAD-mimicking pyridazine group and a 4-(5-(trifluoromethyl)pyrimidin-2-yl)-piperazine that binds in the adenosine subpocket, linked together by an ethoxyethane moiety. Several attempts have been made since to modify different parts of the RBN-2397 molecule scaffold in order to improve its selectivity towards PARP7. The resulting RBN-2397 derivatives – I-1, (S)-XY-05, Compound 18, and Compound 8, all achieve high inhibitory potency against PARP7 in the nanomolar range while also having increased selectivity for PARP7 over other ART family enzymes such as PARP1 and PARP2. KMR-206 is another highly PARP7-specific inhibitor, made by attaching a propynyl group at the C-6 position of the pan-PARP inhibitor Phthal01, both of which are based on a phthalizinone scaffold. Another phthalizinone-based small molecule is the pan-PARP inhibitor thioparib, which can overcome initial or acquired olaparib resistance, and anti-tumor properties of which are directly linked to its ability to inhibit PARP7. Compound 36 belongs to a new series of quinazoline-2,4(1H,3H)-dione derivatives and exhibits excellent potency towards PARP1, PARP2, and PARP7, with PARP7 being the highest. Finally, given the synergistic effect observed between RBN-2397 and pembrolizumab, efforts have been made towards developing B3 and C6 – bifunctional conjugates, which target both PARP7 and PD-L1. (B) Schematic representation of the cyclization strategies used to modify the RBN-2397 scaffold in order to increase PARP7 specificity.

CT26 cells with either inhibitor was used to validate the role of PARP7 in IFN-I signaling in cancer. Head-to-head comparison of RBN-2397 and KMR-206 showed that both inhibitors induce a dose-dependent increase of STAT1 and pSTAT1, whereas no induction was observed for pTBK1 and pIRF3 was only marginally increased. Cell viability assay using NCI-H1373 cells demonstrated that, despite both inhibitors having similar potency against PARP7, KMR-206 was 6-fold less potent in decreasing NCI-H1373 viability and increasing *IFNB1* expression than RBN-2397. This suggests that not just catalytic inhibition, but another mode of action—perhaps PARP "trapping", is behind the immunostimulatory effects of PARP7 inhibitors.

## Pan-PARP inhibitors

In addition to selective PARP7 inhibitors, thioparib and Cpd36 were developed as pan-PARP inhibitors also targeting PARP7 (Fig. 5, Table 3). Thioparib efficiently inhibits PARP1, PARP2 at low nanomolar concentrations and PARP3, TNKS1, TNKS2, PARP6, PARP7, PARP8, PARP11, and PARP1 at higher nanomolar concentrations (Wang et al, 2023). In addition, it exhibits superior potency to current clinically approved PARP inhibitors talazoparib and olaparib, and various cancer cells with initial or acquired olaparib resistance remain sensitive towards thioparib. The superior anti-proliferative properties of thioparib are attributed to its ability to inhibit PARP7 as well, which leads to the induction of the IFN-I response via the STING/TBK1 signaling pathway. Moreover, PARP7 is indispensable for the anti-tumor properties of thioparib, as PARP7 KO MC38 cells treated with thioparib exhibit no tumor growth suppression, whereas PARP1 KO tumors remain sensitive.

In a recent study, Zhou et al synthesized compound 36 (Cpd36)—a quinazoline-dione derivative, which demonstrates nanomolar potency against PARP1/2/7 (Zhou et al, 2023). In BRCA1-deficient breast cancer and PTEN-deficient prostate cancer mouse models Cpd36 was shown to be more effective than olaparib, whereas in mice bearing MC38 colon cancer Cpd36 increases the proportion of cytotoxic CD8+ T cells.

## Bifunctional PARP7/PD-L1 inhibitors

Complex diseases like cancer involve not just one, but many altered pathways. Not just that, but treatment of only one aberrant aspect oftentimes ends in the development of resistance due to inefficient targeting, which is why combinatory therapy, where multiple enzymes or pathways are addressed, yields better results. Given that the synergistic effect of RBN-2397 with an anti-PD-1 antibody was

previously established, developing a small molecule inhibitor, which targets both PARP7 and PD-1/PD-L1 is the next logical step in the search for new and powerful therapeutics. Gao et al were able to synthesize two bifunctional conjugates—B3 and C6—which demonstrated potent activity against both PD-1/PD-L1 and PARP7 (Gao et al, 2024) (Fig. 5, Table 3). B3 had a higher preference for PARP7 than RBN-2397 (IC$_{50}$ = 2.5 nM), whereas C6, albeit with a slightly lower potency for PARP7 (IC$_{50}$ = 7.1 nM), displayed 27-fold lower potency for PARP1. Both novel compounds still demonstrated nanomolar potency against PARP2, similar to RBN-2397. The in vivo anti-tumor activity of B3 and C6 was evaluated in mouse melanoma B16-F10 tumor model and the results showed 5.3-fold higher efficacy than RBN-2397.

## Conclusion

PARP7 regulates metabolism and immune response by acting on AHR and IFN-I signaling pathways. Its aberrant expression in cancer has been linked with proliferation, disease progression, and immune evasion. Initially identified as a target gene of AHR, we now know that PARP7 interacts with many other transcription factors, which regulate various physiological processes and are implicated in cancer, making PARP7 an attractive target for the development of novel therapeutics. Furthermore, PARP7 has emerged as a key negative regulator of the IFN-I signaling—a pathway important not only for innate immunity but also for malignancy. In the framework of cancer, PARP7 exhibits a highly context- and cell-specific role, acting mainly as a tumor promoter. Therefore, it is of utmost importance to identify new disease-relevant PARP7 substrates and to uncover their function in order to piece together the bigger picture behind PARP7-mediated oncogenesis.

In recent years the first PARP7-specific inhibitor was developed—RBN-2397. PARP7 inhibition in different cancer models and in vivo experiments helped with the characterization of PARP7 functions and revealed its therapeutic potential. RBN-2397 exerts its anti-tumor effect mainly by restoring the IFN-I signaling pathway and increasing the recruitment of CD8+ T cells to the tumor site. Currently, multiple RBN-2397 derivatives, as well as inhibitors with a different scaffold, have been synthesized in an attempt to increase the potency and selectivity for PARP7. Given the synergistic effect between RBN-2397 and anti-PD-1 antibodies, bifunctional conjugates targeting both PARP7 and PD-1/PD-L1 have been developed as a novel approach for treating cancer.

In conclusion, further research into PARP7 substrates and how their function is altered upon PARP7-mediated MARylation or its

loss in malignancy could help create better strategies for targeting and treatment of cancers such as breast, ovarian, lung, pancreatic cancer and glioma.

## Graphics

Figure 2 was created with BioRender.com.

### Pending issues

1. Identification of PARP7 substrates is of utmost importance for understanding how PARP7 regulates the IFN-I response and other pathways.
2. Mechanistic understanding of how PARP7 regulates its substrates may help with the design of substrate-specific inhibitors of PARP7.
3. Testing combinatorial effects of PARP7 inhibitors and other therapeutic agents (e.g., radiation, DNA damage response inhibitors, immune checkpoint inhibitors) may provide new treatment options for different cancer types.

## Peer review information

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

## Acknowledgements

This work was supported by the Technopol grant K3-F-730/003-2020 and FWF doc.funds.connect PAIR 'Pre-clinical ion beam therapy' (DFH 13).

## Author contributions

**Katerina Popova**: Conceptualization; Visualization; Writing—original draft; Writing—review and editing. **Johannes Benedum**: Visualization. **Magdalena Engl**: Visualization. **Carola Lütgendorf-Caucig**: Writing—review and editing. **Piero Fossati**: Writing—review and editing. **Joachim Widder**: Writing—review and editing. **Klaus Podar**: Writing—review and editing. **Dea Slade**: Conceptualization; Supervision; Writing—original draft; Writing—review and editing.

## Disclosure and competing interests statement

KIP has received speaker's honoraria from Celgene, Amgen Inc., and Janssen Pharmaceuticals, consultancy fees from Celgene, Takeda, and Janssen Pharmaceuticals, and research support from Roche Pharmaceuticals. The authors declare no competing interests.

