## [Peer Review File · EMBO Molecular Medicine]

PARP7 as a new target for activating anti-tumor immunity in cancer

Dea Slade, Katerina Popova, Johannes Benedum, Magdalena Engl, Carola Lütgendorf-Caucig, Piero Fossati, Joachim Widder, and Klaus Podar

Corresponding author: Dea Slade (dea.slade@meduniwien.ac.at)

Review Timeline:

Submission Date:	25th Nov 24
Editorial Decision:	20th Dec 24
Revision Received:	24th Jan 25
Editorial Decision:	24th Feb 25
Revision Received:	25th Feb 25
Accepted:	3rd Mar 25

Editor: Lise Roth

Transaction Report:

20th Dec 2024

Dear Dr. Slade,

Thank you for the submission of your review to EMBO Molecular Medicine. We have now received feedback from the experts who agreed to evaluate your manuscript.

As you will see from the reports below, they overall found the review interesting and well written. They nevertheless make several suggestions to improve the interest and impact of your work.

We would therefore welcome a revised version of your manuscript that would address these points. Please attach a covering letter giving details of the way in which you have handled each of the points raised by the referees.

- 1/ A .doc formatted version of the manuscript text (including Figure legends and tables)
- 2/ Separate figure files
- 3/ A letter INCLUDING the reviewer's reports and your detailed responses to their comments.
- 4/ A glossary: EMBO Molecular Medicine articles are accompanied by a glossary explaining some of the terms used for laymen.
- 5/ Pending issues: At the end of each article, there is a box highlighting issues that still need further studies and where research efforts should converge (called the Pending issues box).
- 6/A 'disclosure statement and competing interests' statement (<https://www.embopress.org/competing-interests>).

For the figures:

We work with one of our expert scientific illustrators, who will assist with getting the figure to a publication ready state. What we need from you is a draft that accurately illustrates the key scientific concepts that you wish to show.

Please also note the following points:

- If there are certain aspects of your figure draft that are based upon assumptions or where the scientific data remains ambiguous, please add a comment so that we can work with you on an accurate depiction. Please ensure the directionality and nature of interactions is presented accurately.
- If the figure or single panels of the figure have been adapted from a published figure, please add this information to the figure legend (e.g., 'Adapted from...' or 'Based on...').
- Please only re-use figures or parts of a figure if this is essential for understanding the concept communicated. Often a reference to a previous paper will suffice. If the figure contains re-used images or elements of images, please make sure that you have the permission/license to publish it (this also applies to your own previous work, if the journal you published in retains copyright. Certain 'creative commons' open access licenses, such as CC-BY 4.0, allow re-use without additional formal permissions). All re-used material must be explicitly cited.
- If you use an image data base for scientific iconography (e.g., BioRender), please let us know if you have a license that allows for publication in an academic journal. Often authors use misleading iconography for expedience. Please ensure the information shown is scientifically accurate.
- For figures created using a software for editing vector objects like Inkscape, CorelDraw etc., please send the file as a PDF (or SVG, or EPS), PowerPoint or Keynote in which the labels and objects are still editable. For figures created using Adobe Illustrator, please send the Illustrator (.ai) file.

Looking forward to receiving your revised manuscript,

With my best wishes for the holiday season,

Lise Roth

***** Reviewer's comments *****

Referee #1 (Remarks for Author):

The paper by Popova and colleagues attempts to review the biochemical and physiological properties of PARP7. There was a large number of novel studies on PARP7 recently and, indeed, it is advantageous to put these studies into a bigger picture as the review aims to. I am in favor of the manuscript as it fills in a gap in knowledge. Nevertheless, I suggest to review the paper to attempt to provide a large picture. There are chapters where one paper is summarized after the other, but the knowledge is not synthesized (the best example is the inflammation/immunology chapter). Please review findings from the perspective of the pathology, why is something important for the initiation or progression of a disease? What is the role of a physiological process and what is the role of PARP7 in it?

My specific questions are the following:

1. The abstract should be more detailed, should answer the question what does PARP7 do/is important for.
2. The paper is underreferenced. There are multiple statements that are not underscored by referring to the original site of description. Eg.: "The catalytic domain of PARylating PARPs, such as PARP1, PARP2, PARP5a and PARP5b, contains the H-Y-E motif, where histidine and tyrosine are needed for the binding of the NAD⁺ molecule, whereas glutamate is important for the generation of PAR chains. However, despite PARP3 and PARP4 also having this signature motif, they are unable to PARylate their substrates, implying that other residues also play a role in determining the catalytic activity."
3. ER and LXR are nuclear receptors, furthermore, AHR nuclear receptor-like functions. The fact that PARP7 mediated nuclear receptors-mediated transcription could be called out in the review. In the pharmacology section AR (I suspect it's androgen receptor) shows up...
4. Page 4., last paragraph. What is Kyn? Kynurein by chance, or a protein?
5. Please review the abbreviations, e.g. MEF, IFN-I, AR, etc. are not resolved.
6. Please reshape the IFN-I chapter. What I miss is a clear goal where this chapter goes to. Although, I have the same feeling with neuroinjury and cancer as well.
7. I truly like the idea to discuss the involvement of PARP7 in individual cancers separately and not as a big mishmash. However, for the moment the chapter is a listing of facts. I wonder if it would be possible to transfer listings into a table and discuss the available material in a less descriptive way.
8. The maintenance of pluripotency is not restricted to PARP1 and PARP7. Why is pluripotency discussed together with neuroinjury?
9. p9. "Both inhibitors increase endogenous PARP7
10. protein levels, suggesting that its catalytic activity regulates its protein levels (Sanderson et al., 2023)." Its in the sentence refers to what? Furthermore, I do not agree to the sentence directly. Is it possible that these inhibitors influence PARP7 stability or half-life without binding to PARP7?
11. There is a ~10-20 fold difference in the IC₅₀ of RBN-2397 on PARP7 and PARP1/2/12. I would be somewhat more careful when discussing the specificity of that inhibitor, as in an in vivo setting this difference may not provide strong selectivity for PARP7.
12. p9. "Although RBN-2397 shows clear selectivity for PARP7, the IC₅₀ values for PARP1, PARP2 and PARP12 are 37nM, 17nM and 25nM respectively, raising the question as to whether the observed effects are due to PARP7 inhibition alone. RBN-2397-mediated effect on the IFN-I response was comparable in wild-type and PARP1, PARP2 or PARP3 knock-out (KO) cells, confirming its on-target effect." Isn't this contradiction in terms?
13. Please provide a figure for the cyclization of RBN-2397 compound.
14. I may have missed, but the metabolic regulatory/disruptor role of PARP7 (OXPHOS, lipid metabolism, etc.) is not discussed in sufficient detail, but is the quoted in the discussion.
15. For cancer survival/gene expression database screening I suggest to screen multiple databases besides TCGA event if those would rely on the TCGA dataset (though would always complement with further patients). Eg. gepia2, kmplot.com, ualcan.

Referee #2 (Remarks for Author):

General Comments

The manuscript by Slade and colleagues is timely and reviews the current literature on PARP7. It effectively covers the "state of the art" regarding PARP7 functions and inhibitor development, a topic of significant interest considering PARP7's involvement in cancer. However, there is a noticeable overlap with recent reviews, particularly the one by P. Manetsch and M.O. Hottiger (BioEssays, 2024 Nov 6), which limits the novelty of this manuscript.

Below, I provide detailed comments and suggestions aimed at improving clarity and addressing some gaps in the content, as the flow is occasionally hard to follow.

Specific Comments

1. Abstract section (Page 2)

- The authors define PARP7 as a "mono-PARP", which is outdated nomenclature. According to the review by Lüscher et al. ("ADP-ribosyltransferases, an update on function and nomenclature"), this terminology should no longer be used. Please revise this according to the updated nomenclature. I suggest to cite this manuscript, which has been endorsed by the majority of experts in the field.

2. "Enzymatic Properties of PARP7" section (Page 2)

The authors focus solely on proteins as PARP targets, whereas it is widely accepted that nucleic acids are also targets. This aspect should be included for completeness.

3. Missing references:

- "The catalytic domain of PARylating PARPs, such as PARP1, PARP2, PARP5a, and PARP5b, contains the H-Y-E motif..."
- "Despite PARP3 and PARP4 having this signature motif, they are unable to PARylate their substrates..."

Please ensure that appropriate references are provided throughout the text, as some points lack proper citations.

4. "PARP7 Biological Functions and Substrates" section (Page 3)

PARP7 Localization: The statement "PARP7 localizes mainly in the nucleus..." is vague. Does it also show cytoplasmic localization? Please clarify and provide references.

Substrates: References are missing for the statement: "AlphaFold3 revealed a model of PARP7 binding to its known substrates (e.g., tubulin, HIF-1 α , and NF- κ B RelA)." Including details from Ahmed et al., 2015, would enhance this section.

5. IFN-I Suppression (Page 5)

The manuscript states that "PARP7 was suggested to suppress the IFN-I response by ADP-ribosylating TBK1, preventing its activation." Given the conflicting data in the literature, the authors should include relevant studies that present alternative findings and revise "Figure 2" accordingly.

6. "Pluripotency and Neurogenesis" section (Page 8)

This section appears out of context given the manuscript's title and focus. Either remove this section or better contextualize it within the main narrative.

7. "RBN-2397 and Derivatives" section (Page 9)

The section lacks clarity, particularly regarding the inhibitor's specificity. For instance: The manuscript reports that RBN-2397 shows good IC50 values for PARP1, PARP2, and PARP12, raising questions about whether the observed effects are exclusively due to PARP7 inhibition.

On page 10, the authors state that "the anti-tumor immunogenic effects of RBN-2397 can be enhanced in PARP2, 3, 12, and 1/2 KO CT26 tumors," which seems contradictory.

Please revise this section to improve clarity, ensuring it is accessible to a broader audience beyond ADP-ribosylation specialists.

8. Figure 2

o PARP13: This protein is mentioned in the figure but not in the main text or the figure legend. Please address this inconsistency.

o For alpha-tubulin, include the functional readout of PARylation to maintain consistency with other pathways depicted.

o Proteasome Pathways: The figure suggests a connection between the ARH and FRA1 pathways, which is misleading. Modify this for clarity and align the FRA1 pathway with NA sensing signaling, considering previous comments on TBK1 and PARP7.

o Inhibitor Scheme: Grouping inhibitors by selectivity is useful, but including the pathways inhibited by these drugs would enhance the figure's utility.

9. Table

The table lists PARP7 substrates with corresponding references. However, the references are numbered, while the references section does not use numbering. This inconsistency should be corrected.

In summary, the manuscript addresses a critical and timely topic; however, several revisions are necessary to improve its clarity, coherence, and accessibility. To align with the journal's guidelines, the authors may consider to integrate the text with supplemental explanations and context to help non-specialist readers better understand the content. This will ensure that the manuscript meets the journal's criteria of being engaging and informative for a broader audience.

Referee #3 (Remarks for Author):

The manuscript by Popova et al. provides a review of the literature on PARP7's biological functions, with an emphasis on its immunomodulatory role in cancer. This timely review would be useful for researchers developing novel anti-cancer therapies. The manuscript may be accepted in EMBO Molecular Medicine. However, I suggest that the authors implement the following revisions.

The suggested improvements are summarized below.

1. On page 3 in the section on PARP7's biological functions and substrate, the authors state that PARP7 is mainly localised in the nucleus and suggest that the expression of this enzyme in different compartments of the cell is only marginal. It's in apparent contradiction with the localisation of PARP7 depicted in Figure 2, which suggests the abundance of PARP7 in the cytosol. Furthermore, it appears that for the specific processes on which the review particularly focuses, such as the regulation of IFN-I signalling and the inhibition of PARP7, the presence of this enzyme in the cytosol is crucial. This discrepancy needs clarification.
2. Figure 2 contains too much information and is insufficiently connected to the text. I suggest splitting the Figure into two or three separate figures and positioning them close to the text where the corresponding biological processes are discussed. For example, the TBK1 regulation part of the current Figure 2 could be placed in the section on IFN-I regulation, and the part dealing with the inhibitors in the section describing the inhibitory action of those compounds. The latter part should also be clarified. In the introductory part of the section on PARP7 inhibitors a translocation of PARP7 to the nucleus is suggested, but this is not apparent from Figure 2 in its current state.
3. Figure 3C contains no useful information and should be removed.
4. The "PARP7 inhibitors" section requires figures illustrating the structures of the inhibitory compounds mentioned in this section. The authors could then refer to the structures of the inhibitors in the text to enhance the reader's understanding. For example, it is impossible to appreciate the statement "... an optimized inhibitor of the ethyloxyl linkage" on page 10 without visualisation of the structures of the relevant compounds. On page 11, the authors refer to reported inhibitors by their numbers in the cited literature (compound 8, compound 18). This practice does not facilitate the reading of the article, but the problem can be easily solved by including a figure with the structures of the relevant compounds depicted.

We would like to thank all reviewers for taking the time to provide an in-depth critical feedback on our manuscript. We have incorporated all required changes in the current version (highlighted in yellow) and provided a point-by-point response to all comments below.

Referee #1 (Remarks for Author):

The paper by Popova and colleagues attempts to review the biochemical and physiological properties of PARP7. There was a large number of novel studies on PARP7 recently and, indeed, it is advantageous to put these studies into a bigger picture as the review aims to. I am in favor of the manuscript as it fills in a gap in knowledge. Nevertheless, I suggest to review the paper to attempt to provide a large picture. There are chapters where one paper is summarized after the other, but the knowledge is not synthesized (the best example is the inflammation/immunology chapter). Please review findings from the perspective of the pathology, why is something important for the initiation or progression of a disease? What is the role of a physiological process and what is the role of PARP7 in it?

My specific questions are the following:

1. The abstract should be more detailed, should answer the question what does PARP7 do/is important for.

The abstract was expanded to include major PARP7 substrates and functions.

2. The paper is underreferenced. There are multiple statements that are not underscored by referring to the original site of description. Eg.: "The catalytic domain of PARylating PARPs, such as PARP1, PARP2, PARP5a and PARP5b, contains the H-Y-E motif, where histidine and tyrosine are needed for the binding of the NAD⁺ molecule, whereas glutamate is important for the generation of PAR chains. However, despite PARP3 and PARP4 also having this signature motif, they are unable to PARylate their substrates, implying that other residues also play a role in determining the catalytic activity."

The missing references were added.

3. ER and LXR are nuclear receptors, furthermore, AHR nuclear receptor-like functions. The fact that PARP7 mediated nuclear receptors-mediated transcription could be called out in the review. In the pharmacology section AR (I suspect it's androgen receptor) shows up...

We modified this section to highlight PARP7-mediated regulation of nuclear receptors.

4. Page 4., last paragraph. What is Kyn? Kynurein by chance, or a protein?

Yes, it's kynurein, this is now properly defined.

5. Please review the abbreviations, e.g. MEF, IFN-I, AR, etc. are not resolved.

All abbreviations are defined at the first mention in the text.

6. Please reshape the IFN-I chapter. What I miss is a clear goal where this chapter goes to. Although, I have the same feeling with neuroinjury and cancer as well.

These chapters were rewritten to improve clarity.

7. I truly like the idea to discuss the involvement of PARP7 in individual cancers separately and not as a big mishmash. However, for the moment the chapter is a listing of facts. I wonder if it would be possible to transfer listings into a table and discuss the available material in a less descriptive way.

We introduced a table summarizing the links between PARP7 and different cancer types (new Table 2). We decided to keep the text as it provides additional information on the mechanistic aspects that are difficult to summarize in a table.

8. The maintenance of pluripotency is not restricted to PARP1 and PARP7. Why is pluripotency discussed together with neuroinjury?

Pluripotency was discussed in the same chapter as neuroinjury because PARP7 depletion in stem cells leads to accelerated neuronal differentiation, reduced migration and proliferation of neural stem cells and aberrant organization of the cerebral cortex. However, based on the comment from Reviewer 2 that this section appears out of context, we decided to remove it from this manuscript.

9. p9. "Both inhibitors increase endogenous PARP7 protein levels, suggesting that its catalytic activity regulates its protein levels (Sanderson et al., 2023)." Its in the sentence refers to what? Furthermore, I do not agree to the sentence directly. Is it possible that these inhibitors influence PARP7 stability or half-life without binding to PARP7?

We cited the paper (Sanderson et al., 2023) that showed the increase in PARP7 protein levels upon PARP7 inhibition, where they explain these findings as catalytic activity regulating protein levels through ubiquitination and proteasomal degradation. We now added an additional explanation to support this claim: 'PARP7 protein levels are higher for the catalytically inactive mutant and upon proteasome inhibition (Lu et al., 2019) and the ubiquitin ligase HUWE1 only interacts with WT PARP7 (Zhang et al., 2020), suggesting that PARP7 stability is regulated by its catalytic activity in a ubiquitin-dependent manner.'

10. There is a ~10-20 fold difference in the IC50 of RBN-2397 on PARP7 and PARP1/2/12. I would be somewhat more careful when discussing the specificity of that inhibitor, as in an in vivo setting this difference may not provide strong selectivity for PARP7.

p9. "Although RBN-2397 shows clear selectivity for PARP7, the IC50 values for PARP1, PARP2 and PARP12 are 37nM, 17nM and 25nM respectively, raising the question as to whether the observed effects are due to PARP7 inhibition alone. RBN-2397-mediated effect on the IFN-I response was comparable in wild-type and PARP1, PARP2 or PARP3 knock-out (KO) cells, confirming its on-target effect." Isn't this contradiction in terms?

We rephrased this section: 'RBN-2397-mediated effect on the IFN-I response, as assessed by the induction of CXCL10, was comparable in wild-type and PARP1, PARP3 or PARP12 knock-out (KO) CT26 cells, but was reduced in PARP2 KO cells, in accordance with the smallest difference in IC50 between PARP7 and PARP2 (Gozgit et al., 2021).'

11. Please provide a figure for the cyclization of RBN-2397 compound.

This is now included as Figure 5.

13. I may have missed, but the metabolic regulatory/disruptor role of PARP7 (OXPHOS, lipid metabolism, etc.) is not discussed in sufficient detail, but is the quoted in the discussion.

Not much is known about the metabolic functions of PARP7 and this is already included in this paragraph:

'PARP7 was shown to promote or suppress the toxic effects of AHR agonists. The AHR agonist 2,3,7,8-tetrachlorodibenzo-p-dioxin (TCDD) induces the lethal wasting syndrome characterized by suppression of gluconeogenesis (Diani-Moore et al., 2010). PARP7 plays a major role in mediating the effects of TCDD in chick embryo hepatocytes, as it reduces the expression of phosphoenolpyruvate carboxykinase (PEPCK) and glucose-6-phosphatase (G6Pase) – key enzymes of gluconeogenesis. By depleting NAD⁺ levels in the liver, PARP7 reduces the activity of another NAD⁺-dependent enzyme – SIRT1, which is necessary for the activation of the gluconeogenic gene coactivator PGC1 α . Conversely, in knock-out (KO) mouse models, PARP7 was shown to have protective effects by (i) suppressing TCDD-induced hepatosteatosis (fatty liver disease) (Ahmed et al., 2015), and (ii) suppressing the effects of the AHR agonist 3-methylcholanthrene (3MC) on chylous ascites (peritoneal fluid that is rich in triglycerides) (Cho et al, 2019).'

We now added one more sentence: 'As a co-activator of LXR, PARP7 positively regulates SREBP1 and SREBP2, which are responsible for cholesterol and fatty acid import and synthesis (Bindesbøll et al., 2016).'

14. For cancer survival/gene expression database screening I suggest to screen multiple databases besides TCGA event if those would rely on the TCGA dataset (though would always complement with further patients). Eg. gepia2, kmplot.com, ualcan.

Figure 4 now includes KM plots from all these databases.

Referee #2 (Remarks for Author):

General Comments

The manuscript by Slade and colleagues is timely and reviews the current literature on PARP7. It effectively covers the "state of the art" regarding PARP7 functions and inhibitor development, a topic of significant interest considering PARP7's involvement in cancer. However, there is a noticeable overlap with recent reviews, particularly the one by P. Manetsch and M.O. Hottiger (BioEssays, 2024 Nov 6), which limits the novelty of this manuscript.

Below, I provide detailed comments and suggestions aimed at improving clarity and addressing some gaps in the content, as the flow is occasionally hard to follow.

Specific Comments

1. Abstract section (Page 2)

- The authors define PARP7 as a "mono-PARP", which is outdated nomenclature. According to the review by Lüscher et al. ("ADP-ribosyltransferases, an update on function and nomenclature"), this terminology should no longer be used. Please revise this according to the updated nomenclature. I suggest to cite this manuscript, which has been endorsed by the majority of experts in the field. This is now corrected and the reference is included.

2. "Enzymatic Properties of PARP7" section (Page 2)

The authors focus solely on proteins as PARP targets, whereas it is widely accepted that nucleic acids are also targets. This aspect should be included for completeness.

We included a statement about nucleic acids as ART targets: 'Some ARTs can also modify terminal phosphates and bases of nucleic acids (Munnur et al, 2019; Musheev et al, 2022; Talhaoui et al, 2016).'

3. Missing references:

- "The catalytic domain of PARylating PARPs, such as PARP1, PARP2, PARP5a, and PARP5b, contains the H-Y-E motif..."

- "Despite PARP3 and PARP4 having this signature motif, they are unable to PARylate their substrates..."

These references are now added.

Please ensure that appropriate references are provided throughout the text, as some points lack proper citations.

4. "PARP7 Biological Functions and Substrates" section (Page 3)

PARP7 Localization: The statement "PARP7 localizes mainly in the nucleus..." is vague. Does it also show cytoplasmic localization? Please clarify and provide references.

We already included an extended description of PARP7 localization:

PARP7 localizes mainly in the nucleus, which requires the N-terminal NLS region and an intact zinc finger domain CCCH (Gomez et al., 2018) (Fig. 1B). PARP7 localizes also in the cytosol, as observed in HeLa and MCF7 cells (MacPherson et al, 2014; Vyas et al, 2013). Furthermore, following infection with Sindbis virus (SINV), PARP7 accumulates in the cytosol of mouse embryonic fibroblasts (MEFs) (Kozaki et al, 2017). This translocation is facilitated by oxidation of the nucleoporin complex via reactive oxygen species (ROS) from damaged mitochondria, which is mediated by the BCL2 family members BAX and BAK1. PARP7 nuclear localization is crucial for its function as a modulator of nuclear receptor-mediated transcription and IFN-I signaling, while its cytosolic localization may contribute to the regulation of IFN-I signaling and microtubules, which will be described in detail below.

Substrates: References are missing for the statement: "AlphaFold3 revealed a model of PARP7 binding to its known substrates (e.g., tubulin, HIF-1 α , and NF- κ B RelA)." Including details from Ahmed et al., 2015, would enhance this section.

We did the modeling ourselves, so the only reference that is needed here is for AlphaFold3, which is now included (Abramson et al, 2024). We now included Ahmed et al., 2015 as an additional citation showing that PARP7 transactivation domain MARYlates AHR: 'Reciprocally, PARP7 regulates AHR by MARYlating key peptides within its transactivation domain, which leads to its proteolytic degradation and downregulation of AHR target genes (Ahmed et al, 2015; Gomez et al., 2018; Lu et al, 2019; MacPherson et al., 2014) (Table 1).'

5. IFN-I Suppression (Page 5)

The manuscript states that "PARP7 was suggested to suppress the IFN-I response by ADP-ribosylating TBK1, preventing its activation." Given the conflicting data in the literature, the authors should include relevant studies that present alternative findings and revise "Figure 2" accordingly.

We are aware of the conflicting results in the literature, which we addressed by stating: 'In PARP7 KO MEFs infected with the vesicular stomatitis virus (VSV), pTBK1 levels were found to be increased (Yamada et al., 2016). This effect is dependent on the activation of the IFN-I signaling pathway as pTBK1 levels do not differ between PARP7 KO and WT untreated cells (Sanderson et al, 2023; Yamada et al., 2016). However, pTBK1 levels were not increased in PARP7 KO EO771 mouse mammary cancer cells upon induction of the IFN-I signaling with the STING agonist DMXAA and PARP7 inhibition did not increase pTBK1 levels in wild-type cells treated with DMXAA (Rasmussen et al, 2023). PARP7-mediated effects on pTBK1 may therefore be cell line- and treatment-dependent. PARP7 likely has multiple substrates within the IFN-I pathway as IFNB1 induction was weaker in PARP7 KO cells compared to TBK1-depleted cells (Rasmussen et al., 2023).'

We included as part of Figure 2 legend the following statement: 'In mouse embryonic fibroblasts (but not mouse mammary cancer cells) it was shown that PARP7 ADP-ribosylates TBK1, preventing its autophosphorylation and thus its kinase activity, which impairs the IFN-I response.'

6. "Pluripotency and Neurogenesis" section (Page 8)

This section appears out of context given the manuscript's title and focus. Either remove this section or better contextualize it within the main narrative.

We agree that this section is out of context and decided to remove it.

7. "RBN-2397 and Derivatives" section (Page 9)

The section lacks clarity, particularly regarding the inhibitor's specificity. For instance: The manuscript reports that RBN-2397 shows good IC50 values for PARP1, PARP2, and PARP12, raising

questions about whether the observed effects are exclusively due to PARP7 inhibition. On page 10, the authors state that "the anti-tumor immunogenic effects of RBN-2397 can be enhanced in PARP2, 3, 12, and 1/2 KO CT26 tumors," which seems contradictory. Please revise this section to improve clarity, ensuring it is accessible to a broader audience beyond ADP-ribosylation specialists.

These contradictory statements were now clarified:

‘In CT26 tumor-bearing mice, RBN-2397 induced durable tumor regression, which is mediated by CD8 T-cells and enhanced in combination with anti-PD-1 antibodies (Gozgit et al., 2021). Its anti-tumor effects are dependent on the presence of PARP7, STING and TBK1 (Gozgit et al., 2021).’

8. Figure 2

o PARP13: This protein is mentioned in the figure but not in the main text or the figure legend. Please address this inconsistency.

PARP13 is now removed from the figure.

o For alpha-tubulin, include the functional readout of MARYlation to maintain consistency with other pathways depicted.

Alpha-tubulin is now properly depicted in Figure 2.

o Proteasome Pathways: The figure suggests a connection between the ARH and FRA1 pathways, which is misleading. Modify this for clarity and align the FRA1 pathway with NA sensing signaling, considering previous comments on TBK1 and PARP7.

This is now corrected.

o Inhibitor Scheme: Grouping inhibitors by selectivity is useful, but including the pathways inhibited by these drugs would enhance the figure's utility.

Based on Reviewer 1 suggestion, we included a separate figure with PARP7 inhibitors (Fig. 5).

9. Table

The table lists PARP7 substrates with corresponding references. However, the references are numbered, while the references section does not use numbering. This inconsistency should be corrected.

This is now corrected.

In summary, the manuscript addresses a critical and timely topic; however, several revisions are necessary to improve its clarity, coherence, and accessibility. To align with the journal's guidelines, the authors may consider to integrate the text with supplemental explanations and context to help non-specialist readers better understand the content. This will ensure that the manuscript meets the journal's criteria of being engaging and informative for a broader audience.

Referee #3 (Remarks for Author):

The manuscript by Popova et al. provides a review of the literature on PARP7's biological functions, with an emphasis on its immunomodulatory role in cancer. This timely review would be useful for researchers developing novel anti-cancer therapies. The manuscript may be accepted in EMBO Molecular Medicine. However, I suggest that the authors implement the following revisions.

The suggested improvements are summarized below.

1. On page 3 in the section on PARP7's biological functions and substrate, the authors state that PARP7 is mainly localised in the nucleus and suggest that the expression of this enzyme in different compartments of the cell is only marginal. It's in apparent contradiction with the localisation of PARP7 depicted in Figure 2, which suggests the abundance of PARP7 in the cytosol. Furthermore, it appears that for the specific processes on which the review particularly focuses, such as the regulation of IFN-I signalling and the inhibition of PARP7, the presence of this enzyme in the cytosol is crucial. This discrepancy needs clarification.

We clarified this conundrum by stating: 'PARP7 nuclear localization is crucial for its function as a modulator of nuclear receptor-mediated transcription and IFN-I signaling, while its cytosolic localization may contribute to the regulation of IFN-I signaling and microtubules, which will be described in detail below.' Moreover, we removed PARP7 inhibitors from Figure 2 (they were previously placed in the cytosolic part only for space reasons) and made a new figure with their structures.

2. Figure 2 contains too much information and is insufficiently connected to the text. I suggest splitting the Figure into two or three separate figures and positioning them close to the text where the corresponding biological processes are discussed. For example, the TBK1 regulation part of the current Figure 2 could be placed in the section on IFN-I regulation, and the part dealing with the inhibitors in the section describing the inhibitory action of those compounds. The latter part should also be clarified. In the introductory part of the section on PARP7 inhibitors a translocation of PARP7 to the nucleus is suggested, but this is not apparent from Figure 2 in its current state.

As explained above, we 'cleaned up' Figure 2 and placed PARP7 inhibitors in a separate figure. The structure and clarity of this figure is hopefully improved now.

3. Figure 3C contains no useful information and should be removed.

We would like to keep this figure panel as it shows the type and frequency of PARP7 mutations along the protein sequence.

4. The "PARP7 inhibitors" section requires figures illustrating the structures of the inhibitory compounds mentioned in this section. The authors could then refer to the structures of the inhibitors in the text to enhance the reader's understanding. For example, it is impossible to appreciate the statement "... an optimized inhibitor of the ethyloxyl linkage" on page 10 without visualisation of the structures of the relevant compounds. On page 11, the authors refer to reported inhibitors by their numbers in the cited literature (compound 8, compound 18). This practice does not facilitate the reading of the article, but the problem can be easily solved by including a figure with the structures of the relevant compounds depicted.

We included a new figure with PARP7 inhibitors (Fig. 5).

24th Feb 2025

Dear Dr. Slade,

Thank you for the submission of your revised manuscript to EMBO Molecular Medicine. We have now received the feedback from referee #1, who evaluated your revised manuscript, and as you will see below, he/she is satisfied with the revisions and supports publication. I will therefore be able to accept your manuscript once the following editorial issues are addressed:

- Remove the yellow highlighted font and only keep in track changes mode any new modification.
- Please provide up to 5 keywords.
- Author contributions: CRediT has replaced the traditional author contributions section because it offers a systematic machine readable author contributions format that allows for more effective research assessment. Please remove the Authors Contributions from the manuscript and use the free text boxes beneath each contributing author's name in our system to add specific details on the author's contribution.
- Please add the Glossary and Pending issues to the manuscript file.
- Please remove the figures (but not their legends) from the manuscript file. Figures should be called out in the text in the chronological order. Currently, Fig. 3A is called out before Fig.2, please adjust.
- Figure 2: please increase the font size to make the text legible.
- Figure 3E is difficult to read: we would suggest placing it vertically, with bigger circles.
- If you use an image data base for scientific iconography (e.g., BioRender), please let us know if you have a license that allows for publication in an academic journal.

Looking forward to receiving your revised manuscript,

With kind regards,

Lise

Lise Roth, Ph.D.
Senior Editor
EMBO Molecular Medicine

***** Reviewer's comments *****

Referee #1 (Remarks for Author):

None.

The authors addressed the remaining editorial changes.

3rd Mar 2025

Dear Dr. Slade,

Thank you for submitting your revised files. I am pleased to inform you that your manuscript is accepted for publication and is now being sent to our publisher to be included in the next available issue of EMBO Molecular Medicine.

Your manuscript will be processed for publication by EMBO Press. It will be copy edited and you will receive page proofs prior to publication. Please note that you will be contacted by Springer Nature Author Services to complete licensing information.

If you have any questions, please do not hesitate to contact the Editorial Office.

Thank you for your contribution to EMBO Molecular Medicine!

With kind regards,

Lise Roth
